# The structure and evolutionary diversity of the fungal E3-binding protein

Bjoern O. Forsberg ⬡ [1,2✉]

The pyruvate dehydrogenase complex (PDC) is a central metabolic enzyme in all living cells composed majorly of E1, E2, and E3. Tight coupling of their reactions makes each component essential, so that any loss impacts oxidative metabolism pathologically. E3 retention is mediated by the E3-binding protein (E3BP), which is here resolved within the PDC core from *N.crassa*, resolved to 3.2Å. Fungal and mammalian E3BP are shown to be orthologs, arguing E3BP as a broadly eukaryotic gene. Fungal E3BP architectures predicted from sequence data and computational models further bridge the evolutionary distance between *N.crassa* and humans, and suggest discriminants for E3-specificity. This is confirmed by similarities in their respective E3-binding domains, where an interaction previously not described is also predicted. This provides evolutionary parallels for a crucial interaction human metabolism, an interaction specific to fungi that can be targeted, and an example of protein evolution following gene neofunctionalization.

[1] Department of Physiology and Pharmacology, Karolinska Institutet, Biomedicum, Solnavägen 9, 171 77 Stockholm, Sweden. [2] Present address: Division of Structural Biology, Wellcome Centre for Human Genetics, University of Oxford, OX3 7BN Oxford, UK. ✉email: bjorn.forsberg@ki.se

The 2-oxoacid dehydrogenase complexes are a class of protein complexes with acyl-transferase activity in metabolism and amino acid synthesis[1]. They co-localize 3 catalytic components (E1-3) that act in sequence upon a substrate-carrying lipoyl domain(LD) that is flexibly linked to E2. This couples the reactions[2] and increases their overall rate through so-called substrate channeling. The pyruvate dehydrogenase complex (PDC) is the 2-oxoacid complex responsible for the bulk production of acetyl-coenzyme A (CoA) in all cells, crucial to oxidative phosphorylation, but also histone acetylation in the nucleus[3]. Dysfunction and regulation of the PDC is consequently implicated in many disorders characterized by altered metabolism, including cancer[4–7]. Genetic abnormalities also impact the PDC[8,9] and more recently it has also been recognized that the PDC is a source of reactive oxygen species, which impacts signaling cascades and the metabolic state of the cell[10].

The arrangement of 2-oxoacid dehydrogenase complexes vary[11], but are invariably built around the C-terminal catalytic transacetylase domain (CTD) of E2. The CTD forms trimers that arrange into higher order assemblies. Further, E2 has two N-terminal domains separated by flexible linking regions. The N-terminal lipoyl domain (LD) shuttles the pyruvate-derived acetyl group via a covalent lipoamide modification. The central domain of E2 is a peripheral-subunit binding domain (PSBD) that tethers E1 to the CTD-composed core (Fig. 1a). It has been previously established that the mammalian PDC employs an E2 paralog to recruit E3 via a specialized PSBD named E3-binding protein (E3BP)[12–14]. Mammalian E3BP has identical domain topology to E2, but is catalytically inactive[15], and has been concluded to substitute one or more E2 core units in an unknown fashion. Several models of the mammalian core E2:E3BP ratio have been suggested and studied[16–22], but no consensus has been reached, nor is the reason for the catalytic inactivity of E3BP known. Like mammals, fungi also utilize E3BP to recruit E3[21,23–25]. Unlike mammals however, fungal E3BP binds to the interior of the E2 core assembly instead of substituting core components[25–28] (Fig. 1b). Mammalian and fungal E3BP have been treated as separate entities in literature, perhaps mainly due to the fundamental difference in their C-terminal domain and its mode of binding. The fungal E3BP has been denoted "protein X" (PX). Both E3BP and PX have influenced similarity-based automatic annotations, leading to confusion in current databases. To distinguish it from the CTD of E2 and reflect their shared features, the C-terminal domain of E3BP or PX is here named as the core-binding domain (CBD), regardless of taxonomic origin.

In this study, the molecular structure of the CTD:CBD sub-complex of the fungal PDC from *N.crassa* was determined using cryo-EM to 3.2Å resolution. It unambiguously demonstrates the determinants for their interaction, the mode of E3BP oligomerization interior to the E2 core, and the homology of the CBD and CTD. It is shown that neo-functionalization of E3BP from an ancestral E2 gene likely predates the reduction of the fungal CBD, which has subsequently diverged structurally from that of e.g. mammals, while preserving essential E3BP-functionality. Fungal and mammalian E3BP are thus determined as orthologs. This suggests that the E3BP function may be much more ubiquitous to eukaryotes than previously thought. In line with this, variations on the CBD are found in fungi outside the Pezizomycota (Pez) subphylum that *N.crassa* belongs to. Examples are evident where the CBD appears more similar to its ancestral CTD fold, but there are also examples where the CDB is even further reduced than in *N.crassa*. In addition, the observed binding mode of fungal CBD is compatible with a much more complete E2-like fold, suggesting that core-internal localization is possible in non-fungal species. Sequence analysis of their respective PSBD corroborate the orthology of fungal and mammalian E3BP, and permits an extensive bioinformatical analysis of its specificity for E3. In addition, the flexible linker connecting the *N.crassa* CBD and PSBD shows a conservation pattern that suggests relevance for interaction with the lipoyl domain (LD) as it interacts with E3.

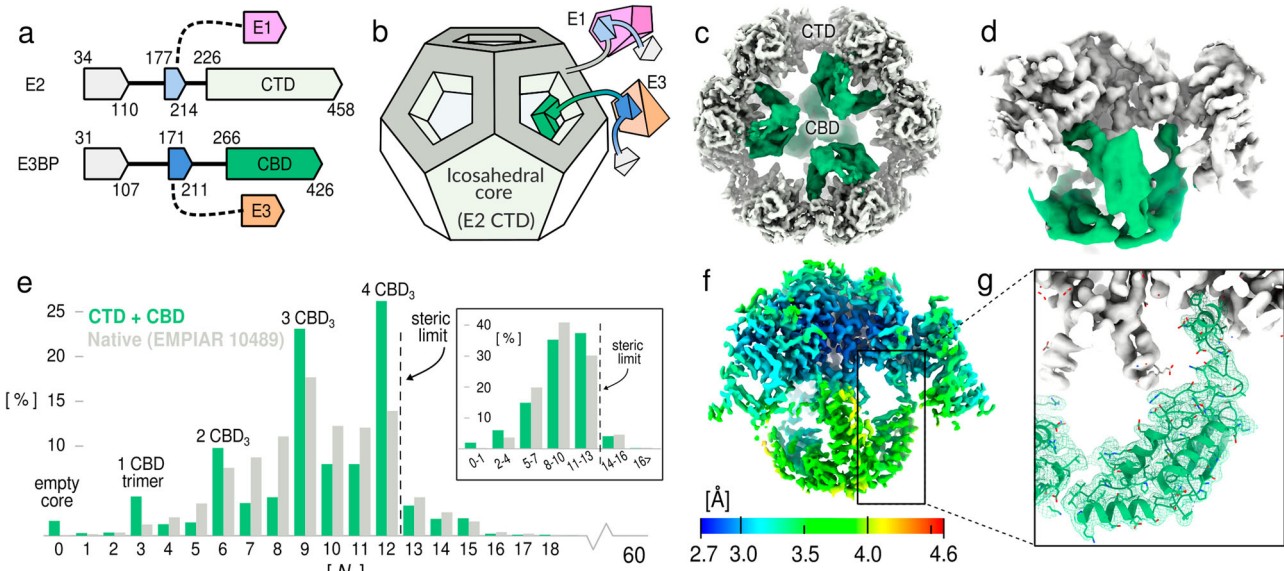

**Fig. 1 Cryo-EM reconstructions of *N.crassa* E2:E3BP-CBD sub-complex. a** Domain topology of E2 and E3BP, and **b** The overall arrangement of these proteins in the fungal PDC. Adapted from Forsberg et al.[26]. **c** The recombinant CTD:CBD sub-complex reconstructed with tetrahedral symmetry to comply with a maximal CBD:CTD ratio, here colored as E3BP CBD (green) interior and E2 CTD core complex (gray). **d** Local view of a single CBD trimer interior to the CTD core. **e** The fraction of PDC cores with N out of 60 expanded local regions classified as being occupied by a CBD trimer. The inset show the same histogram grouped across the indicated bins, which signifies the most likely number of trimers. For instance, either 8, 9, or 10 particles identified as trimers is taken to most likely to signify 3 trimers, and are grouped to indicate PDC cores containing 3 CBD trimers. Data available as Supplementary Data 1. **f** The improved resolution following symmetry expansion, signal subtraction, and further alignment and classification of the local region shown in panel **d**, colored by estimated local resolution. **g** The atomic model of the CBD monomer in the improved density.

These findings comprehensively furthers our understanding of the separate recruitment of E3 to the PDC and its range of variation among eukaryotes, and elaborates on modalities in this crucial metabolic complex that deserve further study.

## Results

**Structure of the E3BP CBD**. To determine the interactions of the fungal E3BP:E2 subcomplex, their CBD and CTD respectively (Fig. 1a, b) were recombinantly co-expressed and examined by cryo-EM (Supplementary Fig. 1). Tetrahedral symmetry was used for the preliminary reconstruction (Fig. 1c), and 3-fold symmetric sub-complexes were computationally isolated by symmetry expansion and signal subtraction (Fig. 1d). CBD-occupied E2-trimers were selected by classification, and mapped back to the cores particles from which they were extracted, which permitted analysis of how many CBD trimers were found within each core particle (Fig. 1e). The majority of E2 core particles had 3-4 CBD trimers assigned, whereas only 4% were assigned 5 or more CBD-trimers. This agrees well with a limit of 4 due to steric considerations[26], and these are thus attributed to false positive identification within the margin of error of data classification. Identified CBD-occupied E2 trimers were then further classified and re-aligned under C3-symmetry. This resulted in elevated CBD occupancy as evidenced by reconstructed intensity, and reduced conformational heterogeneity (flexibility). The final reconstruction to 3.2 Å (Fig. 1f, Supplementary Fig. 1) utilized less than one CBD-trimer per well-aligned PDC core particle, and permits the CBD from *N.crassa* E3BP to be built de novo (Fig. 1g). The analysis was repeated using native PDC data EMPIAR-10489[29] (Supplementary Fig. 2), which found a highly similar CBD-trimer distribution. Increased uncertainty in both classified proportions and attained were however noted due to the higher overall noise in this data, to the extent that it can only quantifiably corroborate the existence and predominance of the CBD trimer in native samples compared to the recombinant expression, and conversely cannot confirm specific interactions observed in the latter.

The core fold of the *N.crassa* CBD consists of four helices and two flanking beta-strands (Fig. 1g), inherited from an ancestral CTD. The CBD of *N.crassa* has a trimeric interface with a symmetry axis that coincides with that of a core E2 CTD-trimer (Fig. 2a). The CBD trimer interface has evolved from the dimeric interface that core CTD trimers form to extend to larger assemblies. It is hydrophobic in nature, formed largely from the C-terminal end of the first CBD helix (residues L285, I289, and V291), as well as it's C-terminus (Fig. 2b). Further to this, residues V281, L293 F292 and F298 form an extended hydrophobic core that is shielded from above by P420 and V424, stabilizing the CBD C-terminus. The C-terminus if further potentially stabilized by R301, which displays clearly resolved side-chain density (Supplementary Fig. 3) and is highly conserved across Pez fungi. The C-terminal residues of the CBD in *N.crassa* is also noteworthy, being Leu/Val/Ile in 95% of examined Pez sequences, and Arg only in Neurospora.

To validate the role of R301 directly, molecular dynamics (MD) simulations were conducted of the CBD trimer bound to a partial PDC core (see methods and Supplementary Fig. 4). Even during comparatively short simulations, R301 shows frequent intermolecular interactions with the C-terminal carboxyl (Supplementary Fig. 5c), which appears to stabilize the CBD trimer by domain swapping. Intra-molecular contact between R425 and E304 is also recurring and likely stabilizes the CBD C-terminus (Supplementary Fig. 5c). These interactions are conceivable in a hypothetical dimeric CBD oligomer. Further simulations were therefore conducted, where one of three CBD monomers was omitted. The same interactions were observed (Supplementary Fig. 5d), but the overall flexibility of the dimer was increased due to the reduced constraints, reducing oligomer stability. Based on this evidence, R301 is likely to stabilize the CBD-terminus. The trimer should also be favorable over a dimer configuration, whereas the latter cannot be refuted based on present observations.

Beyond the core fold, the CBD contains a coil region (S338-S348) that lines the helices adjacent to the beta-sheet where Y342 shields the hydrophobic core of the CBD (Fig. 2c). One might expect F346 to be tighter against I299, F393 and L275, but the density instead supports a solvent-exposed hydrophobic pocket (Fig 2d). Hence, S338-S348 appears to be a partially stabilized coil region. This coil region bridges two regions that protrude from the CBD core fold, each containing a small and highly conserved motif denoted M2 (T319-L328) and M3 (D360-A367) respectively[26]. M2 is a clearly resolved short helix connected to the CBD by structured loops, and is responsible for binding to E2 (Fig. 2e). M3 is flanked by regions of predicted disorder (residues 347-389), none of which is resolved in the current reconstruction (Fig. 2c). MD does not reveal frequent or patterned interactions for the M3-containing loop, but corroborates its disordered nature (Supplementary Fig. 6). Simulations of the monomeric CBD in solution displays similar variability in disordered region but retained overall stability of the core fold. The termini of the M3 disordered region which remain structured are within a 5Å of each other, and positioned near the 5-fold pseudo-symmetry axis of the E2 core assembly, placing it approximately equidistant from each of the 5 closest E2 trimers (Supplementary Fig. 7).

**E2-E3BP interactions**. As shown previously[26,28], the CTD-trimers form a homomeric dimer interface and hydrophobic pocket to which the CBD of fungal E3BP binds (Fig. 2e). The present reconstruction shows unambiguously that M2 is the major binding interface of the E2 core. In *N.crassa*, M2 is composed of a amphipathic helix which is conserved within Pez, and which likely extends to Saccharomycetes (Sac) (Fig. 3a, b). A conserved Phe (*N.crassa* F324) appears crucial for binding, as predicted in *C.thermophilum*[28]. Additionally, K266 from both participating CTD monomers interact with M2 residues N325 and Q326 from the CBD (Fig. 2e). K263 of the central E2 trimer is also interacts with CBD residue D321. As CBD binding is oriented within the strictly symmetric pocket, binding of monomeric CBD is likely meta-stable. CBD trimerization provides avidity of binding in this scenario, enforcing oriented M2 binding. M2 remains bound in all conducted simulations. The M3 motif is not resolved in the present reconstruction, being situated in the disordered region of *N.crassa* CBD connecting F346 and E391 (45aa) (Fig. 2c). M3 however recapitulates the general properties of M2 (Fig. 3c). M3 might thus be a secondary binding motif. Residual density is observed in the CTD binding pockets where trimeric CBD is sterically impermissible (Supplementary Fig. 7b), but as it cannot be clearly resolved auxiliary M3 binding should be considered possible but unsubstantiated. Conducted simulations also did not indicate a preference for the M3 region to approach or favorably interact with available binding pockets. The observed density could alternatively be M2 of meta-stable monomeric CBD, that does not participate in canonical CBD trimers. The universal conservation of M3 thus remains unexplained.

Flexibility in the CBD is evident from data processing (Supplementary Fig. 8). This is validated by the conducted simulations of monomeric and trimeric CBD, which display comparable ranges of flexibility of the M2-containing loop that

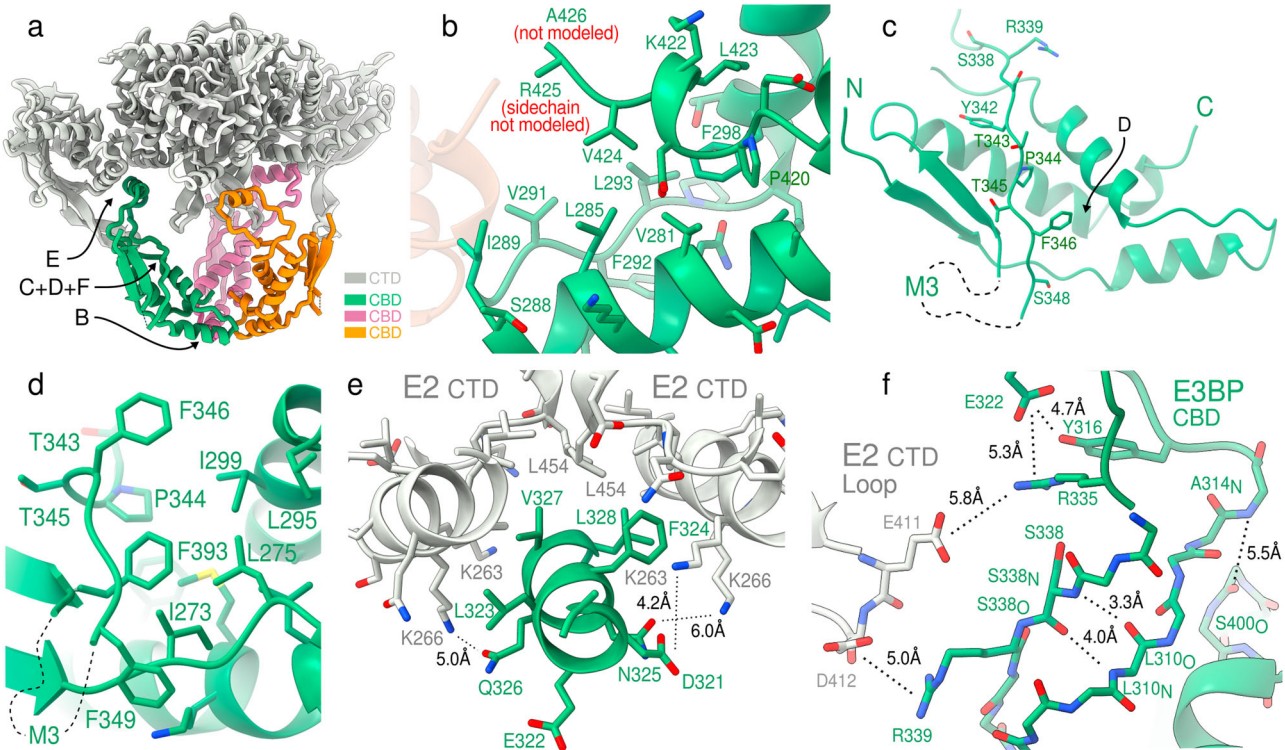

**Fig. 2 Molecular interactions and homology of fungal E3BP. a** The overall architecture of the E2 CTD trimer, its E2 neighbors interacting through a twofold-symmetric interface, and the core-interior CBD trimer with individually colored monomers. Arrows indicate regions highlighted in subsequent panels. **b** The CBD homomeric trimer interface is exclusively hydrophobic, barring R425, which is unresolved and atypical of the fungal CBD. **c** A partially hydrophobic strand incompletely shields the CBD hydrophobic core from solvent. The arrow indicates the region of the view in panel **e**. **d** Detailed view of the hydrophobic core of the CBD shown in panel **c**. **e** The binding motif is centered around a hydrophobic patch on the CTD complementary to CBD F324, and charge complementarity to CTD lysines K266 and K263. **f** Numerous potential salt bridges and electrostatic interactions are mediated by the E2 CTD core-internal loop (white). Interactions supported by molecular dynamics simulations (see Supplementary Fig. 5) are indicated in dashed lines. Annotated distances are instantaneous distances of the deposited model PDB-7r5m, and not equilibrium or binding distances observed in simulations.

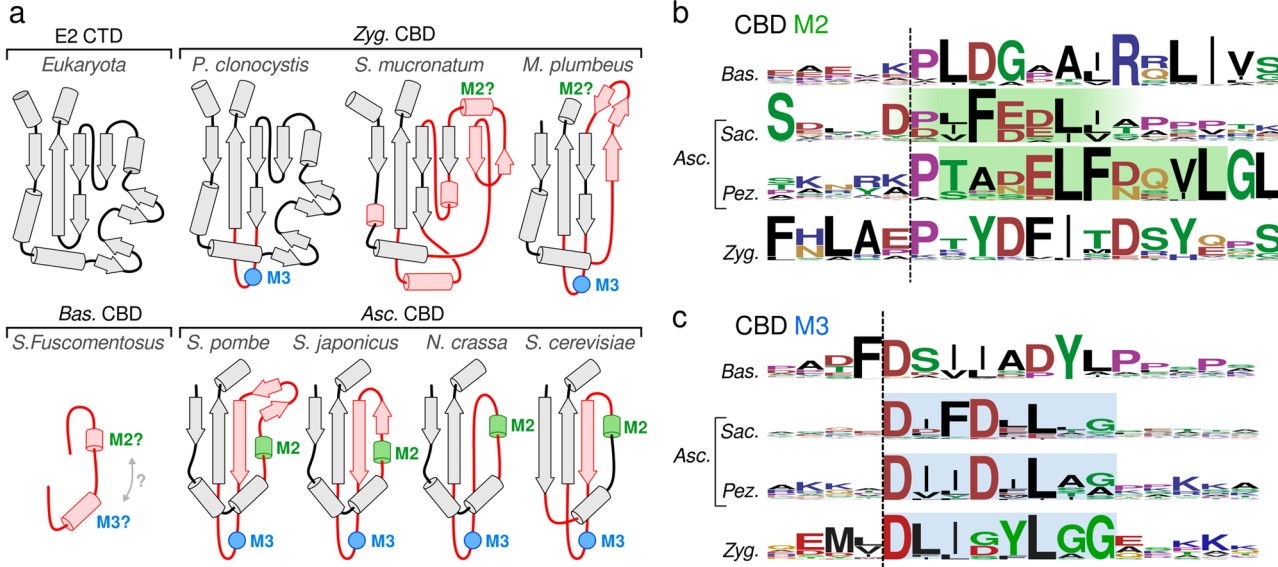

**Fig. 3 Topology of fungal E3BP. a** The E2 CTD fold topology schematic of eukaryotic E2 CTD, compared to that of the CBD of E3BP in several fungal species, grouped by phylum. Topologies are sequence-based predictions apart from that of *N.crassa*, and display a broad variation in evolution from the ancestral CTD of a duplicated E2 gene. Substantial differences to the CTD is indicated in red, and conservation motifs are also indicated as M2 and M3 in Ascomycota (Asc). Inferred M2 and M3 sequence motifs in Zygomyceta (Zyg) and Basidiomycota (Bas) are indicated, with varying confidence. **b, c** The sequence logo of M2 and M3 regions are given for each fungal CBD phylum, and confident assignment is highlighted as based on inferred sequence similarity and conservation compared to *N.crassa*.

anchors the CBD to the PDC core (Supplementary Fig. 6a, b). Since such flexibility results in variable asymmetry of the CBD trimer, no asymmetric features could be confidently established using cryo-EM data. There are however interactions between the CBD and the core-internal beta-strand loop of the CTD that stabilize the CBD trimer. Molecular dynamics support the interactions of CTD residues E411 and D412 with CBD residues R335 and R339 respectively (Fig. 2f). The CTD loop supplying these potential interactions is universal to 2-oxoacid dehydrogenase complexes, but varies in length and composition across species and E2 substrate-specificity. No role has been suggested for this loop, but here shows electrostatic stabilization to a core-internal binding partner. The same loop of adjacent CTD trimers within the PDC core assembly are also proximal to the CBD, and can be observed in the periphery of the localized reconstruction, near the center of the PDC core. R335 also appears to stabilize the M2-containing loops through E322 and Y316 (Fig. 2f, Supplementary Fig. 5b).

**Animal and fungal E3BP are orthologs**. E3BP and E2 are established homologs in fungi based on sequence data[30] despite their comparatively large topological differences. Computational modeling of *C. Thermophilum* E3BP also predicted that its CBD was a partial E2 fold[28]. The reconstruction of the *N.crassa* CBD confirms that fungal E3BP and E2 are indeed structurally similar. Orthology of mammal and fungal E3BP is probable, but unsubstantiated. Its occurrence in fungi has e.g. not been demonstrated outside Ascomycota (Asc). To investigate if fungal and mammalian E3BP are diverged orthologs as opposed to arisen by convergent evolution, the characteristics of E3BP were generalized and queried against existing sequence databases. E3BP was identified as proteins displaying i) a domain topology consistent with E2 homology, ii) an E3-specific PSBD (IPR004167, IPR036625, SSF47005), iii) a catalytically inactive acetyltransferase-homologous CBD, and/or iv) an M3-like motif of unknown function. Three distinct forms of fungal E3BP were identified. First, the E3BP of Basidiomycota (Bas) is inferred from criteria i) and iv) and appears reduced to two short stretches of predicted helical structure, which resemble M2 and/or M3 motif (Fig. 3b, c). The Bas CTD binding pocket is similar to that of Asc, with conservation of residues equivalent to *N.crassa* K263 and K266. This suggests required avidity through oligomerization as in Asc, without a steric limit to E3BP binding. Alternatively, dual binding motifs per CBD monomer may increase binding affinity. The Bas CBD also holds a short conserved Y/F-L/F-DGL$\phi$-motif ($\phi$ = hydrophobic). Second, the CBD of Asc

matches all characteristics i)–iv), since it includes those studied here and previously in *S.cerevisiae*[25], *N.crassa*[26], and *C.thermophilum*[28]. Its CBD is a CTD-derived fold that is broadly similar to that of *N.crassa* (Fig. 3a). There are however variations in Taphrinomycotina and Saccharomycotina that will require structural confirmation to verify e.g. CBD oligomeric state. M2 and M3 can however be unambiguously assigned throughout Asc, based on similarity to *N.crassa*. Third, the CBD of Zygomyceta (Zyg) is also match all criteria i)–iv), but is far less consistent than Asc. Sequence data for Zyg is also more sparse than for Asc, but frequently display a CBD that is more similar to its CTD ancestor (Fig. 3a). Some species display a locus and motif similar to M3, but that of M2 is less certain. Combined with the more CTD-like fold, it is unclear if the Zyg CBD will form core-interior binding or core-substitution. Under hypothetical core-substitution, its core-internal loop is much longer than that of any mammalian counterpart (typically 50aa). Hence, Zyg E3BP clearly shows distinctively fungal properties, but is quite dissimilar from any Asc species. The tripartite division (Bas/Asc/Zyg) of fungi (Supplementary Fig. 9) outlines the extent of fungal variation of the CBD as detected by the established search criteria. The range of variation and shared features validate their mutual origin and recapitulate the evolutionary history of the reduced CBD in Asc. It therefore seems likely that fungal and animal E3BP are indeed orthologs, with a shared evolutionary origin, and that E3BP is a eukaryotic gene.

**PSBD specificity**. The PSBD which anchors peripheral PDC components was omitted from the recombinantly produced sample used here, and was consequently not reconstructed or directly visualized. However, to further validate the orthology of fungal and mammalian E3BP, and to seek discriminants E1/E3 for specificity using the range of fungal sequences found by the generalized E3BP criteria, E2 and E3BP protein sequences of fungal phyla and animals were analyzed and compared within the PSBD. Multiple-sequence alignments (MSAs) of each (sub)phylum was established, and analyzed in the context of several existing structures. To further aid sequence analysis, alphafold2[31] was utilized to establish multimer-folded structural models of human and *N.crassa* PSBDs from E2 and E3BP, interacting with E1 or E3.

Previous sequence analysis posed that I157 of human E3BP provides specificity for E3[32], contrasted against R383 in the human E2 (Fig. 4a, b, position 38). The present analysis confirms

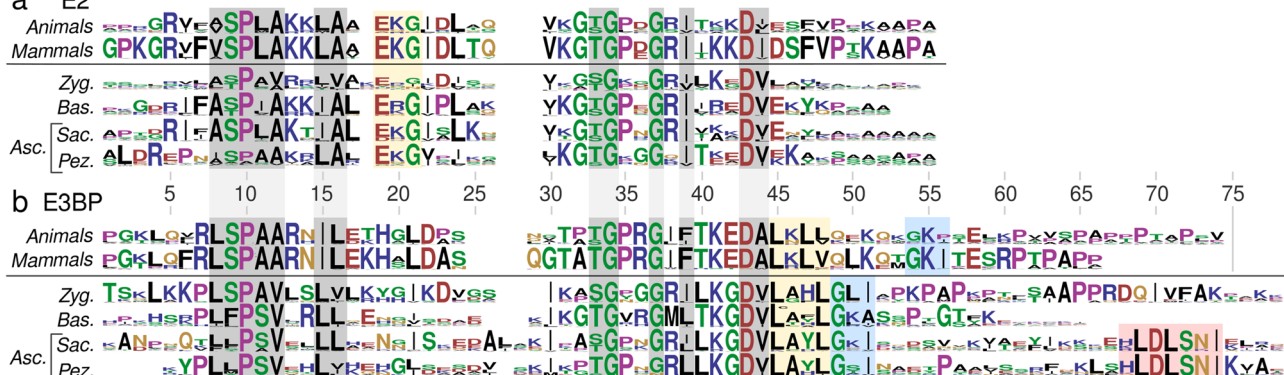

**Fig. 4 PSBD sequence logos of E2 and E3BP. a** PSBD sequence alignment pertaining to E2p (pyruvate dehydrogenase transacetylase) from animals, mammals, and fungal phyla. **b** Alignment as in **a**, pertaining to E3BP. Regions are highlighted as universal to any PSBD (gray), discriminating E2p from E3BP in all examined groups (yellow), PSBD helix 2 termination motif in E3BP (blue), and predicted as extended PSBD E3-binding motif (red). The latter is lead by residue H231 in *N.crassa* (cf. Supplementary Fig. 10).

that human I157 position as distinguishing of E2 from that of E3BP in animals, but the limited sequence variation within animals makes its significance questionable. In further evidence against I157 as directly mediating E3 specificity, fungal species deviate from this pattern. Cross-phylum analysis instead reveals that a single arginine on either side of the immediately preceding glycine is universally preserved. This pattern extends to animal sequences, where E3BP has a conserved arginine (Fig. 4a, position 38, human:R155). Hence, I157 alone is unlikely to supply specificity. Alternatively, three consistently distinguishing sequence motifs that are common to both animal and fungal species are found, which have not been previously noted. The first resides in the C-terminal end of the first helix of the E2 PSBD. (Fig. 4a, position 19-21). This polar motif has consensus sequence 'EKG', and is least prominent in Zyg. Similar residues are frequently found in E3BP, but conservation is only found in E2. The second notable sequence pattern is instead conserved in E3BP but not E2, and consists a hydrophobic motif in the C-terminal end of the second helix (Fig. 4b, position 45-48), following a universally conserved Asp (Fig. 4b, position 43, human:D215, N.crassa:D206). Here, hydrophobic residues and leucine in particular are clearly preferred in E3BP. As part of this motif, a universal leucine (Fig. 4b, position 45, human:L217, N.crassa:L208) is exposed to solvent without any apparent shielding, as is a conserved hydrophobic reside 3 residues downstream (Fig. 4b, position 48, human:V220, N.crassa:L211). Like the polar E2 motif, it unambiguously discriminates the E3BP PSBD from that of E2 across both fungi and animals. The third pattern is lead by a universally conserved glycine that terminates the second helix in the E3BP PSBD. This termination motif has 'GxI' as consensus sequence (Fig. 4b, position 49-51, 54-56 in animals). Predicted models of the binding of the E3BP PSBD to E3 indicate the termination motif may result in different conformations comparing human and N.crassa. Whereas human I228 is predicted to fold back onto the PSBD itself (Supplementary Fig. 10a), N.crassa I214 instead approaches the previously mentioned and highly conserved L208. The termination motif has not been resolved structurally, as 2F5Z[33] includes residues up to E230 but can only resolve residues until T225. No available complexed structure includes the termination motif, whereas 2F60[33] finds residues up to E230 to form a continuous helix in the absence of E3 binding.

All three aforementioned motifs discriminate E2 from E3BP across both fungi and animals, and are thus in clear support of a shared origin and E3BP orthology across eukaryotes. Curiously however, none of these motifs contribute to interactions in the canonical binding interface(Supplementary Fig. 10c, d). Despite the clear conservation of these motifs, present models and evidence cannot substantiate a rationale for their involvement in E3 specificity. One may additionally observe a fourth motif (Fig. 4b, position 68-74) that is specific to Asc, separated from the canonical PSBD by a stretch of low conservation. To examine if this motif offers a further rationale to E3 specificity, the N.crassa E3-PSBD complex predicted by alphaFold was considered (Supplementary Fig. 10b). In this prediction, the region of low conservation from N215 to S230 (Fig. 4b, position 51-67) forms a third and amphipathic helix that lines one E3 monomer, extending towards its substrate pocket (Supplementary Fig. 10b). The conserved motif following it is consistently and confidently predicted as a coil of alternating hydrophobic residues that line the E3 substrate pocket (Supplementary Fig. 10b). The same structure, interactions, or confidence cannot be attributed to a human model, where the linking region is notably shorter and has a much higher proline content (Fig. 4b). These additional interactions are likely specific to the E3BP-E3 interaction and will contribute to specificity. Interactions beyond the canonical

interface may thus contribute to E3 specificity through the three aforementioned conservation motifs that discriminate E2 from E3BP across eukaryotes.

## Discussion

The cryo-EM reconstruction of the CBD from N.crassa E3BP inside the PDC core here reveals its oligomeric form and specific interactions. This permits a confident analysis of variations in the CBD across fungi, which substantiates its orthology to human E3BP. This is further corroborated by sequence motifs in the PSBD that discriminate E2 from E3BP in both fungi and animals. Given that E3BP in fungi and mammals is thus concluded to have descended from the same ancestral gene of their last common ancestor, all species subsequently diverged from this common ancestor could be expected to utilize an E3BP as part of its PDC. E3BP should thus be considered a eukaryotic gene, and annotations as "pyruvate dehydrogenase protein X component" should reasonably be universally changed to E3BP, due to their shared genetic origin.

Given its early occurrence in metabolic evolution, it is pertinent to ask why ancestral E3BP neo-functionalization from an E2 gene was favorable. Naively, recruitment of both E1 and E3 by a shared PSBD as in bacteria implies that their binding ratio is entirely determined by their relative affinity and availability. Utilization of E2 and E3BP as presently discussed decouples the recruitment of peripheral components E1 and E3. This offers a further rationale as to why fungi evolved to add rather than substitute PDC components. Through CBD addition (as opposed to substitution), the fungal PDC maintains a fixed capacity to recruit E1 through the E2 PSBD. Moreover, the 30 binding sites for E3BP impose a fixed E3 capacity as well, which in N.crassa is reduced to 12 through steric restraints imposed by the CBD trimer. The fungal PDC thus combines both decoupled and fixed stoichiometry of E1 and E3 recruitment. Less drastic changes to a core-substituting CBD might similarly dictate the assembly stoichiometry through heteromeric interfaces or decreased stability, as is suggested in the human PDC. This however remains an unproven hypothesis, whereas the fungal PDC is proven to utilize mechanisms to confine the component ratios.

What drove CBD fold reduction to that seen in e.g. N.crassa? First, substantial reduction of the CBD must lead to inability to substitute E2 CTD core subunits. Consequently, CBD fold reduction in fungi likely post-dates a mode of core-addition such as M2. Once established, fold reduction may have been necessary to permit entry into the E2 core interior. As a corollary question, could the full ancestral CBD oligomerize interior to the PDC? Curiously, the full E2 CTD can in fact be superposed on the fungal CBD trimer pose without major clashes. Remarkably, such a core-interior trimer would not even impose any further steric restraint to E3BP binding stoichiometry compared to that of e.g. N.crassa - 12 CBD monomers could still be accommodated (Supplementary Fig. 11). In such an arrangement, beta-loop of the core-internalized CBD monomers would also extend towards each other and offer domain-swapping interactions. Of note, the same loop in the CBD evolved to hold the M3 motif. It would thus be highly interesting to observe the core-interior arrangement of a CBD which is more akin to its CTD ancestor, such as e.g. S.mucronatum or C.reveresa. Thus, while CBD fold reduction likely favors core-internalization, it may not be necessary. The variations in CBD size and components implied by available fungal sequences could then simply correlate with germline mutation rate or environmental factors, but may also reflect the positive enforcement of binding avidity and binding stoichiometry provided through steric occlusion of the PDC interior. There is thus a clear motivation for both the CBD addition model

and CBD fold reduction, while neither is strictly necessary. The addition model also rationalizes the catalytic inactivation of the fungal CBD, whereas it is far less clear why E3BP is catalytically inactive in the proposed substitution model of mammals. This interesting parallel remains to be clarified by future research.

A question that similarly remains unanswered, regards the function of the M3 motif. It is contained within a region of predicted disorder, and is not resolved by the present reconstruction. It is however remarkably conserved throughout fungi. Is M3 an alternate binding motif? The oriented binding of M2 in the symmetric pocket indicates reversible binding. In *N.crassa*, the CBD oligomer provides avidity, but one can also imagine that M3 provides additive affinity. Pez M3 is however more similar to Zyg M3 than to its own M2, which indicates that M3 is conserved by binding to an interface that is different to that of M2 binding. Moreover, previous findings clearly indicate that M3 alone is insufficient to enrich E2 cores by affinity purification[26]. Nevertheless, residual density is observed in the regions of E2 where E3BP would bind were it not sterically prohibited (Supplementary Fig. 7), which could then be due to M3. Can the M3 loop reach these interfaces? The 44-residue loop begins and terminates 45-50Å away from the 3 closest E2-dimer interfaces that are not occupied by E3BP trimer M2. A fully extended protein can reach as far as 3.4 Å per residue. Within the M3-containing loop, only 12 residues precede it and 25 residues follow it. The M3 motif therefore has a reach of only 40–45 Å, unless the CBD partially unfold to lend it further reach. It is thus possible but unlikely that M3 binds to unoccupied dimeric CTD interfaces. The present reconstruction also offers some evidence against this, as one would expect a maximally extended such structure to be less variable and thus resolved. There is no evidence of this. Additionally, such a narrow reach would result in oriented binding and one might similarly expect relatively well-resolved density, which is not observed. Taken together, this indicates that M3 binds weakly or non-specifically to the CTD pocket in the present reconstruction, whereas its physiological target is something else.

Is there any evidence for M3 function in sequence data? In other species of fungi, such as Sac, the equivalent of M3 frequently contains a conserved phenylalanine, distinguishing it from Pez and making it more similar to Pez M2 (Fig. 3b, c). In Sac, the CBD also lacks fundamental structural elements of Pez, and thus it has not been directly observed to form trimers. Hence, the aforementioned additive affinity of M2 and M3 might be more reasonable in Sac than oligomer-mediated avidity as in Pez. Additionally, the trimeric form of *N.crassa* CBD provides a steric limit to 12 bound copies of E3BP per core assembly. In the absence of such steric restraints, additive affinity may similarly enforce a limit of 15 bound copies of E3BP in Sac. Of note, Sac PDC has previously been observed to contain 12 E3BP[24], which instead implies that this too provides steric occlusion by trimerisation. M3 might thus be responsible for binding E3BP to something other than PDC E2, such as another E2 dehydrogenase complex. It may also interact with some core-internal substrate or co-factor such as coenzyme-A. This is substantiated by the observation that the CBD of fungal species of Zyg may well be core-substituting as proposed for mammals, but still contain an extraordinarily long core-internal loop with a motif reminiscent of M3 in the same structural element. Clearly, the purpose of M3 is an interesting but outstanding question.

Finally, the present analysis suggests notable similarities in the PSBD of fungal and animal E3BP, and their mutual differences to that of E2. This solidifies the orthology of fungal and animal E3BP, and suggests discriminants for E3-binding. All established motifs that distinguish the PSBD of E2 from that of E3BP face away from the canonical binding interface. Consequently, an altered PSBD binding pose would be required to argue their direct responsibility for binding specificity for either E1 or E3. This is certainly possible, however the available data shows overwhelming support for the established mode of binding. First, crystallographic structures of human E3BP-E3 complex[32–34] and the E2-E1 complex from a bacterial species[35,36] suggest a singular binding interface and pose of the PSBD, regardless of taxonomy or specificity for E1 or E3. This is corroborated by further studies of bacterial PDC, where the PSBD of E2 is not selective towards E1 or E3[37]. In agreement with this, all predicted PSBD structures recapitulate this overall structure and binding motif. It thus seems unlikely that the fungal PSBD binds in a different pose. Reinforcing this notion, the predicted model of the human E3-E3BP established by colabFold as part of the present work does not depart fundamentally from either 1ZY8[32], or 2F5Z[33]. Instead, the established motifs may offer an opportunity for secondary or even tertiary binding. In such an event, the functionality may also be something other than E1/E3 specificity. An example of this is provided by the structural element presently predicted in the E3-PSBD complex of *N.crassa* (Supplementary Fig. 10b). This feature is unique to Asc, and while it does not implicate the E3 specificity through the polar or hydrophobic motifs within the PSBD, it serves as an example of a possibly important interaction interface that has so far eluded our understanding. Future research will be necessary to elaborate on the polar, hydrophobic, and termination motifs within the PSBD of both E2 and E3BP. In further reference to the predicted structural element and interaction of the extended Asc PSBD (*N.crassa* H231-K241), one might speculate that it affects the binding the lipoyl domain to E3. If so, it would potentially differentiate the two available binding pockets available on each E3 dimer, as only one PSBD binds each such dimer. The implications of this observation are interesting, but will need experimental confirmation. Regardless, the universal conservation of this motif within Asc exemplifies that the flexible linkers of 2-oxoacid dehydrogenases are not as passive as previously assumed. In the absence of more specific alternate hypotheses, present evidence indicates that it augments binding and/or specificity for E3 in Asc. Similar conservation is not found outside Asc. Its conservation is highly interesting in itself, as it constitutes substantial differences between the fungal and human E3BP, while one still finds its PSBD and CBD to contain highly conserved and defining features that cannot currently be assigned a function. Further investigation will be necessary to clarify these interesting correlatives and discrepancies.

## Methods

**Sample preparation and data acquisition**. No new data was acquired, as this work utilized data acquired for[26]. The data acquisition methods and parameters are nonetheless described for completeness. Data acquisition parameters are described in Table 1 and Supplementary Table 1.

Sample was prepared as previously[26]. Briefly, bacterial co-expression of *N.crassa* E2 (residues 225-458, uniprot:P20285) and his-tagged E3BP (residues 261-426, uniprot: Q7RWS2) utilized the petDuet dual-expression vector. The vector was amplified using Escherichia coli DH5-α, and expressed in Escherichia coli Rosetta2 (DE3). For expression, cells were grown in terrific broth at 37 °C and 180 r.p.m. until OD reached 0.5, then induced by 1mM final IPTG. Cells were harvested 3 h post induction. Cells were pelleted and re-suspended in 50 mM Tris pH7.5, 0.5 M NaCl, 20 mM imidazole, then lysed by high-pressure homogenization. Intact cells and debris were pelleted, and the supernatant collected. Ni-NTA agarose slurry was added and incubation under agitation proceeded for at least 30 min. Isolation and washing of Ni-NTA was performed by gravity-flow column, and eluted with 300 mM imidazole. Imidazole was exchanged by size-exclusion chromatography (SEC) on a GE Superose 6-increase 3.2/100, and spin-concentrated.

Grids for cryo-EM were prepared by glow discharge in a Pelco easiGlow. 3 ul of sample was applied to 300-mesh 1.2/1.3 quantifoil grid and vitrified in a FEI Vitrobot mark IV, following 30 s wait, 2 s blot, and 2 s additional wait before plunging. 100% humidity and 4C was maintained prior to plunging. 4063 Micrographs were collected on an 300 kV FEI Krios with a Gatan K2-GIF in counting mode, at a nominal magnification of 165k (0.86 Å/px). At total dose of

**Table 1 Reconstructions. tE2/PX261 referes to the reconstituted subcomplex of co-expressed components.**

| Dataset | tE2/PX261 | Native (EMPIAR-10489) |
|---|---|---|
| Sub-particles | 792,477 | 604,402 |
| EMDB identifier | 14,331 | 16,884 |
| Symmetry (order) | C3 | C3 |
| Sharpening B-factor | −133 | −278 |
| Resolution [Å](FSC @ 0.143) | 3.2 | 4.1 |
| PDB identifier | 7R5M | 8OHS |
| Chains built | 6 E2 + 3 E3BP | 6 E2 + 3 E3BP |
| Bond RMSD | | |
| Length [Å] (#>4$\sigma$) | 0.006 (0) | 0.009 (0) |
| Angles [°] (#>4$\sigma$) | 0.951 (0) | 1.312 (0) |
| Isotropic ADP (min/max/mean) | 0.8/30.8/96.8 | 0.8/25.1/92.6 |
| Molprobity score | 1.17 | 1.33 |
| Clash score | 1.64 | 2.36 |
| Ramachandran [% O/A/F] | 0.0/4.6/95.4 | 0.0/4.5/95.5 |
| Rotamer outliers [%] | 0.8 | 0.2 |
| Model-map FSC [Å] (0/0.143/0.5) | | |
| Masked | | |
| Unmasked | 3.0/3.2/3.8 | 3.6/3.9/4.3 |

31.4 e/Å² was fractionated across 32 frames over 4 s. Grid screening and optimization, as well as data collection was conducted at the Swedish National Cryo-EM Facility at SciLifeLab, Stockholm University and Umeå University.

**Data processing**. In order to resolve fungal E3BP better than previously possible, single-particle cryo-EM data was reprocessed as previously described[26] in RELION[38]. Previous processing applied icosahedral and/or tetrahedral symmetry. In order to attain a higher-resolution reconstruction, a procedure essentially identical to that of localized reconstruction[39] was employed, utilizing symmetry expansion and classification of asymmetric units. The overall procedure is visualized in Supplementary Fig. 1, and details regarding symmetry expansion are provided in Supplementary Note 1. First, PDC core particles were aligned and reconstructed under icosahedral symmetry. The ab-initio algorithm in RELION was used to initialize the optimization, in order to minimize reference bias[38]. Following 2D-classification to remove contamination and poorly picked particles, refinement under icosahedral symmetry, CTF-refinement and polishing were iterated to refine per-particle defocus and global beam-tilt. PDC core particles were next reconstructed by tetrahedral symmetry to align the reconstruction with an orientation compatible with subsequent C3-symmetrical reconstruction. The tetrahedral reconstruction was next the basis for symmetry-expansion, which aligned each E2 trimer of every icosahedral core particle to the same reconstructed E2-trimer. This results in a 60-fold increase in the particle number. A mask that covers the C3-centered E2 trimer was used to subtract complementary density, and the subtracted images were re-sized and re-centered on the C3-centered E2 trimer, to permit additional alignment and classification of E2-trimers pertaining to every original icosahedral particle under C3 symmetry. The distribution of CBD trimers was next mapped back to the original particles, which permitted a tentative analysis of how many CBD trimers were bound to each icosahedral E2 core particle, as shown in Fig. 1e. More details are provided in supporting methods and Supplementary Fig. 1.

Analysis of native samples followed a very similar procedure, with a few exceptions. First, no classification was conducted before consensus refinement as basis for symmetry expansion, since pre-existing particles from EMPIAR-10489 were used. Second, the increased noise level made any re-alignment following symmetry expansion intractable. Positive selection of CBD trimers was thus based on symmetry-expanded consensus alignments only, and any subsequent classification failed to further improve class homogeneity or reconstructed resolution. The deposited map EMD-16884 was therefore reverted to visualize the full extent of the PDC core in which the CBD trimer was identified at the highest possible resolution.

**Model building**. The coordinates of E2 previously published (PDB:6ZLO) were used as a starting point for the modeling of the E2 CTD monomer participating in the symmetry-aligned CTD trimer. An additional CTD monomer was placed to complete the two-fold symmetric homomeric CTD interface. This complex was threefold symmetrized to complete the model of the PDC core CTD present in the reconstruction. Residues 265-346 and 390-425 of *N.crassa* E3BP were then build de novo using coot[40], against the full map sharpened by DeepEMhancer[41]. As the

focused cryo-EM reconstruction following subtraction and re-centering did not encompass the E2 monomer adjacent to the central E2 monomer completely, atoms not supported were next removed. The structure was refined in Phenix[42] against the full EMD-14331 map (post-processed by global B-factor sharpening), and deposited as PDB-7R5M.

The above model was the basis of chain-wise rigid body fitting against the native PDC reconstruction EMD-16884. Partial CTD-chaind were complemented by NCS-mapping, and each chain was next refined in coot[40] with added default chain-wise restraints within 5nm to conserve overall stereo-chemistry. Clashes and other minor issues were resolved manually in coot, and the model deposited as PDB-8OHS. Details of structure refinment and validation statistics are given in Table 1. Figures were made using ChimeraX[43] and PyMOL[44].

**Bioinformatics**. BLAST[45] was utilized to search for E3BP homologs. Jalview[46] was used to manage sequence data, and aligned using the clustal[47] alignment algorithm. Sequence sets were pruned manually, to omit sequences which did not inform on the sought features. It should therefore be noted that possibly erroneous annotations or extremely divergent sequence data may not be included. No filtering for sequence redundancy was performed, but transcripts annotated as 'partial' were removed. For animal sequences, E3BP was discriminated from E2 based on catalytic inactivity as inferred by the absence of a histidine in its canonical transferase triad. For fungal species, sequences were discriminated as E3BP based on mutual similarity in the CBD and the established criteria (see text and methods). Prediction of protein structures and complexes utilized ColabFold[48], using MMseqs2[49]. All modeled sequences are summarized in Supplementary Table 2.

**Molecular dynamics simulations**. A molecular model and simulation of the full PDC core would not be possible to sample sufficiently by atomic simulation. Instead, a fragment of the PDC core was built. The molecular model 7R5M deposited in the present work was the basis of simulations. First, the CBD was complemented with the disordered M3-containing loop as predicted by alphaFold. Further, the PDC core fragment was extended by completing each of the three neighboring CTD trimers, as depicted in Supplementary Fig. 4b. Additionally, the 2 CBD-binding interfaces each such neighbor could supply was completed by adding further fragments of the CTD. All subsequent simulations utilized absolute position restraints of 2 kJ/mol on C-alpha atoms of the CTD fragments and one full CTD monomer of each neighboring CTD trimer. This mimics the soft restraint on global dynamics imposed by the PDC core structure. The model was placed in a triclinic box with dimensions 18.34 nm, 18.54 nm, and 13.51 nm, and box angles 90.0, 90.0, and 60.0 degrees (Supplementary Fig. 4c). TIP3P solvent was added and substituted with Na and Cl ions to a final concentration of 150 mM, and neutralizing overall system charge. This system contained 395'000 atoms. The system was equilibrated by steepest descent to a tolerance of 1000.0 kJ/mol/nm. Subsequently, NVT-ensemble equilibration was conducted over 100 ps using C-alpha position restraints, at 300K (with velocity generation). NPT-ensemble equilibration was finally conducted over 1 ns using C-alpha position restraints, at 300K, and isotropic pressure coupling. This served as a starting point for replica simulations. Each replica repeated NVT- and NPT-equilibration following individual velocity (re) generation. Production simulation used semi-isotropic pressure coupling and 2 kJ/mol position restrains on C-alpha atoms of the protein chains indicated in Supplementary Fig. 4b. CBD trimer and CBD dimer simulations bound to the PDC core fragment was simulated in 8 replicas for at least 100 ns each. For simulations of monomeric CBD in solution, the same procedure was used, within a smaller simulation box, and no position-restrained were applied. Monomeric CBD was simulated in 23 replicas for 100 ns each. All parameter files and input files necessary to reproduce the simulations are provided at Zenodo 7801353[50]. All simulations and preparation utilized the AMBER-99SB force-field[51] in GROMACS 2018.2[52], and VMD[53]. All simulations are summarized in Supplementary Table 3.

**Reporting summary**. Further information on research design is available in the Nature Portfolio Reporting Summary linked to this article.

## Data availability

The atomic model of the recombinantly produced E2-E3BP complex was deposited in the Protein Data Bank as 7R5M and the corresponding map based on the reconstituted E2+E3BP complex as EMD-14331. The EMDB entry provides the full map, as well as the half-maps, mask, and map sharpened by DeepEMhancer[41]. The atomic model of the native E2-E3BP subcomplex determined using data from EMPIAR-10489[29] was deposited as PDB accession code 8OHS and the corresponding map deposited as EMD-16884. All established MSAs, predicted protein models and simulation source files are freely available at Zenodo 7801353[50]. Source data underlying Fig. 1e is provided in Supplementary Data 1.

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

## Acknowledgements

The cryo-EM data were collected at the Swedish national cryo-EM facility, staffed by M. Carroni, J. M. de la Rosa Trevin, J. Conrad, and S. Fleischmann. The manuscript drew benefit from critical evaluation by Pranav Shah. The work was funded by the Swedish research council grant 2020-06413. The computational aspects of this research were also supported by the Wellcome Trust Core Award Grant Number 203141/Z/16/Z and the NIHR Oxford BRC.

## Author contributions

B.F. Collected and analyzed data, and wrote the article.

## Funding

## Competing interests

The author declares no competing interests.
