## [Peer Review File · Communications Biology]

Reviewers' comments:

Reviewer #1 (Remarks to the Author):

The entitled manuscript "The structure and evolutionary diversity of the fungal E3-binding protein" by Bjoern Forsberg was targeted to determine the atomic-resolution cryo-EM structures of the pyruvate dehydrogenase complex. The author has successfully calculated high-resolution cryo-EM structures PDC. This is a well-designed cryo-EM study. However, there are several problems with the methodology. Many results, how these were represented, look extremely premature. I found many unclear findings; Data could have been represented better to support their conclusion. My comments or questions are listed below:

1. In the method section, Cryo-EM data processing details are confusing, and the authors did not explain anything properly in the material method section. It is tough to follow the process. Also, how do authors get 18,379,620 particles from 306,327 using 60x symmetry-expansion. Please elaborate and explain it properly in the method section of data processing. It will be better if the author shows the proper 3D classification.
2. The author has started the results section with "CBD-occupied E2-trimers were selected by classification, and their distribution across PDC core particles analyzed. 4% of CBD trimers were assigned to E2 core particles with 5 or90% of PDC core particles had at least one CBD-trimer bound. Thus, identified CBD-occupied E2 trimers were then further classified and aligned." However, there are no proper figures which could elucidate the results properly? From where does the author determine 4% or 90%. From figures or results it is not clear. The author mentioned 3.2Å resolution, but no proper figures and readers have to search for the FSC curve.
3. The author should mark bad particles in his 2D class averages properly. Otherwise, delete that panel in figure S4. No need to show bad particle or 2D class averages. Most of the time, we do not show bad particles after 2D classifications.
4. Did the author try C1 symmetry? If the author applied C1 symmetry, does the model identical to this current model?
5. How did the authors generate the initial model? It is important to show it properly.
6. Figure 1C, E3BP CBD (green) region of CTD:CBD sub-complex looks extremely low resolution. It will be better if the authors show the 3D model at a different orientation. This will help us to analyze the quality of the 3D model. Also, the author should fit the atomic model into the EM density map to show the overall agreement between the EM map and the atomic model.
7. The atomic model 7R5M builds based on the cryo-EM map E2-E3BP complex (EMDB 14331). However, several amino acids (Ser303-Arg310, Ala226-Val247, and so on) do not fit in the EM map at all. So how do authors build the atomic model of these regions when EM density maps are not visible at all. This is one major concern I have, and authors should rebuild the atomic model. Authors may argue that they built an atomic model based on enhanced.mrc map. However, in case of enhanced.mrc, volume threshold abruptly ended. This is only possible when researchers use a very tight mask or build any mrc model from pdb structure. Additionally, atomic model 7R5M can fit into enhanced.mrc at 0.0337 threshold, however, several extra densities are visible near the EM core map, and the author did not try to explain it properly. I think the author should pay more attention to cryo-EM data processing and model building.
8. These are attributed to false-positive identification within the margin of error of data classification. 90% of PDC core particles had at least one CBD-trimer bound. How do authors determine that "the 90% of PDC core particles had at least one CBD-trimer bound"?
9. "One would expect F346 and possibly L385 to pack against I299 and F393,connected to the CBD by structured loops, and is responsible for binding to E2 (Fig. 2B)." From Figure 2B, none of these are clear. Better representation of figures is highly recommended.
10. "R301 is highly conserved across Pez fungi and stabilizes the C-terminal end of the same CBD monomer." Is there any biochemical evidence about R301 that it is stabilized C-terminal end? Did the author perform any mutation studies to validate the importance of residues L285, I289, V291, P420,

L423, and V424 in the CBD helix? The author should perform a mutation study of Y342 to validate the interaction with E401 and R404. Additionally, the author should clearly mark M2 and M3 regions in the figure. It is extremely difficult to follow which is M2 and M3. Additionally, the authors predicted that K332 interacts with E409, K336 with T414, and so on. However, there is no biochemical evidence provided about these interactions. Is there any published evidence about these interactions and the importance of these amino acid residues?

11. Authors should pay more attention to writing the methods like Model Building, Data Processing, Computational Structure Prediction, Phylogenetic analysis, and so on.

12. Figure S4 legend is very confusing. Like -Sym flag and IcosaCompatWithDefaultTetra.Sym? Why it is important of this type of figure legend. Generally, we don't write all these details in figure legends.

Reviewer #2 (Remarks to the Author):

In this article, the Author presents the 3.2 Å resolution cryo-EM structure of the fungal E3-binding protein (E3BP) in complex with the PDC core, both from *N. crassa*. The measured structural data is complemented by modeling of the PSBD domain and its interactions with the PDC E1 and E3 proteins. Another finding presented in the paper is related to the homology and evolution of eukaryotic and fungal E3BP-like proteins, for which the Author concluded that these are likely orthologs.

The figures are well formatted in general and the only recommendation from my part is for Figure 1. In this figure, the domain topology is presented for both E2 and E3BP, but it would help the readers if amino acid numbering would also be included. Also a mention that the PSBD here was only modeled should be included.

The description of the methods used is suitable for the topic, but for those not familiar with all the expression systems, a mention that the Rosetta bacterial strain is actually *E. coli* might be welcome.

I find the presented work to be novel and suitable for publication. It also contains important conclusions for this field.

Reviewer #3 (Remarks to the Author):

The article entitled "The structure and evolutionary diversity of the fungal E3-binding protein" by Forsberg reports the structure of the *N. crassa* recombinantly expressed C-terminal catalytic transacetylase domain of E2 (CTD) and C-terminal core binding domain of E3BP (CBD) domains utilizing cryoEM at gold-standard 3.2 Å resolution. Based on that and on a broad sequence analysis the author attempts an orthology analysis comparing fungal and mammalian E3BP. Novel findings within the same context are introduced here by the author as the conservation and specificity of the PSBD of E3BP for E3 is also analyzed although it was not part of the structural study. Finally, the author runs alphaFold to predict binding regions for the E3.

As an overall impression, the manuscript is a continuation of the previously published wonderful manuscript of the author (Forsberg et al, Nat Com 2020) that allows more accurate localization of E3BP domain arrangement in respect to the E2 core domains of PDC and a focus on the interacting residues. This submitted manuscript is a purely computational work where data of the previous paper were re-analyzed. The validity of the rest of the data presented here though relies mostly on sequence alignments and predictions without adding experimental verification. Especially the PSBD binding motifs computational prediction could have been validated by an experimental system. Nevertheless, the data presented could enrich current knowledge in PDC and are worth being communicated. The author might find useful the following major and minor points:

Major points

1. Parts of Figure 1 are already shown in the author's previous nature comm paper so I suggest to cite

it. The Nature comm paper also included the PDC from *N. crassa* purified, and the images are available in the EMPIAR by the author. It would be perfectly matching to this manuscript if the claims from the coexpressed E2/E3BP (which were also performed for yeast in 1997) are substantiated in this manuscript as well. This means that image analysis and cryo-EM reconstructions along with 3D models of E2 and E3BP are retrieved from *N. crassa* PDC using his current cryo-EM reconstruction as template and compare to further understand key points raised in the manuscript. Even if achieved resolution would be lower, stoichiometries, potential interfaces etc would add a lot to the current draft.

2. Figure 2 - Lack of PDB validation – depositing more structures from cryo-EM maps calculated here are required. The author does not provide the EMDB/PDB validation report along with the exact deposited PDB model/ EMDB cryo-EM map to assess the quality of the model and its fit in the density – only a table is provided, with e.g. clashscore of low quality (must be better than 2). Currently, due to the lack of these exact data, I cannot comment on the presented data related to cryo-EM. If work from major point (1) is done, then the equivalent validations and models should be provided in the next round of revisions.

3. Figures 3 and 4. Topology analysis of E3BP CBD is not properly citing available literature. Literature related to the E3BP is currently limited in the submitted manuscript and its similarity to E2 is, e.g., known for decades – sequence comparisons and domain comparisons are indeed known. Therefore, the author should better review the PDC literature comparing E2 and E3BP to substantiate or down-tone novelty claims currently present in the manuscript.

4. Comparisons of CBDs. The results of the currently interesting computational analysis of E2/E3BP PSBDs are already discussed in e.g. the Patel book 1996. Even mutational studies have been performed probing bindings of E1 and E3, and even works related to the linker regions and their lengths have thoroughly investigated this. It is suggested that either (some) experimental validation of (some) AlphaFold-based computational models is performed, or the author moves these data to the discussion section with proper referencing. Currently, the manuscript includes experimental data only for the E3BP trimer which were previously collected and included in the Nat Comm 2020 paper. If this is not the case, of course detailed methods and supp should be provided with validations for these new constructs, preparations etc.

5. Please correct typos and inconsistencies in the manuscript (some are mentioned in the minor points) because it was difficult for me to keep track.

Minor points

1. Line 14-15: author mentions that “E3 retention is mediated by the E3-binding protein (E3BP), which has not previously been clearly resolved within the PDC.” which is not true as it has been previously shown by the author himself, in Kyrilis et al Cell reports 2021 (incorrectly cited), in Tüting et al, and later on in Skalidis et al (not cited here). Proteomic, activity, western blot, and cryo-EM reconstruction data have supported this retention. Please revise also claims in Line 45.

2. Line 59: the word “either” maybe should be removed or sentence rephrased?

3. Line 73: typo of word “identification”

4. Line 83: residues P420 and L423 are not properly annotated in Figure 2D. maybe the author could change the illustration to show the full interface? or mention that these residues are not shown. The author can also show the complete CBD and its fit in the high-resolution density in a systematic manner, also by rotating the illustration to show the flanking beta strands as well.

5. Line 94: I think numbering is wrong as it should be 347-389 and not 289, please check.

6. Line 100: the author here stretches a bit the data available as he claims that ONLY M2 is the domain that binds the E2 but the M3 is not clearly resolved thus cannot be excluded as an interacting interface, it would be nice to comment on this, especially when the disordered linker is not included.

7. Line 101: typo“an” amphipathic helix... instead of “a”

8. Lines 109-111: very hypothetical in my opinion, please down-tone the claims.

9. Line 119: Residues K403_CBD-D412-CTD are wrong according to the figure- maybe the author means R425_CBD-D412-CTD?

10. Line 125: the author cites Tüting et al, 2021 manuscript here but does not include it in the comparison of eg Figure S6, nor the model organism.

11. Line 144: words "unambiguously" and "throughout" have typos
12. Line 148-149: sentence that starts with "Were...." I think should be rephrased, maybe it is "Where...?"
13. Line 151-152: author mentions that there are some search criteria in Figure S6 that I don't see. These criteria are not clear to me. Fig S6 shows phylogenetic relationship of fungal species
14. Figure S2 is a bit difficult for me to understand, there are typos in the legend, and in general can you please optimize it? The distances that are presented are not described in the legend.
15. Line 152-154: I think the author here stretches out a bit the available data to make this conclusion, and also reports the obvious of the eukaryotic origin of E3BP.
16. Line 160: word "analyzed" has a typo
17. Line 197: word "specific" typo
18. Line 198: words "with this" mistyped
19. Line 237: is it Fig. S6 or it should be Fig. S7? Please double-check citations to Figures and Tables of the manuscript because I might have missed something.
20. Line 245: "...for the both the..." rephrase
21. Line 263: "Pez M3 is however more similar to Zyg M3 that to its own M2...." Is there a typo here?
22. Line 269: word "Within" has a typo
23. Line 298: ...the PSBD..." space missing
24. Line 307: typo instead of "to", "co is written
25. Line 315 and 324: Celsius miss the degree symbol?
26. Line 376: "s" missing from word "protein" as should be plural
27. Line 520: delete word "the" after "...hydrophobicity that stabilizes..."
28. Line 530: "...symmetry axis" There is a typo here and further on the same line it is either "additional" or "addition of"
29. Line 531: "from utilizing ny of the remaining 3 binding sites that are most proximal to both G347 and V392." Typo
30. Line 532: typo of word "either"
31. In general in the text and also in Supp. Fig S2 the author claims these residual density in the binding pocket. If it was E3BP then also more densities of the E3BP would be visible????maybe the author could explain if he saw more than this except the disordered parts.
32. Line 573: A, B and C are missing from the Fig. S5 even if mentioned in the legend
33. Line 577: word "if" is a typo?
34. Supp Fig S4, please put scale bar in the cryo-EM image

Response to referees

Structure and evolutionary diversity of the fungal E3-binding protein

Bjoern O. Forsberg

COMMSBIO-22-1397-A

2022-12-08

Reviewer comments are in **black**. They have been numbered for easy referencing.

Response is shown in **blue**.

Reviewer #1 (Remarks to the Author):

The entitled manuscript "The structure and evolutionary diversity of the fungal E3-binding protein" by Bjoern Forsberg was targeted to determine the atomic-resolution cryo-EM structures of the pyruvate dehydrogenase complex. The author has successfully calculated high-resolution cryo-EM structures PDC. This is a well-designed cryo-EM study. However, there are several problems with the methodology. Many results, how these were represented, look extremely premature. I found many unclear findings; Data could have been represented better to support their conclusion. My comments or questions are listed below:

1. In the method section, Cryo-EM data processing details are confusing, and the authors did not explain anything properly in the material method section. It is tough to follow the process. Also, how do authors get 18,379,620 particles from 306,327 using 60x symmetry-expansion. Please elaborate and explain it properly in the method section of data processing. It will be better if the author shows the proper 3D classification.

When one uses symmetry expansion, each particle is given multiple alignment parameters, one for every symmetry-equivalent view. Hence, the 306,327 particles expanded by 60-fold symmetry results in 18,379,620 alignment parameters. Subtraction is then performed in each new alignment, leading to the same number of individual (subtracted) particles. This has been clarified in the manuscript methods: "*The tetrahedral reconstruction was next the basis for symmetry-expansion, which aligned each E2 trimer of every icosahedral core particle to the same reconstructed E2-trimer. This results in a 60-fold increase in the particle number.*" The data classification shown in Figure S4 is now clarified (now in Figure S1), adding visual representations of the data selection, within a flowchart.

2. The author has started the results section with "CBD-occupied E2-trimers were selected by classification, and their distribution across PDC core particles analyzed. 4% of CBD trimers were assigned to E2 core particles with 5 or90% of PDC core particles had at least one CBD-trimer bound. Thus, identified CBD-occupied E2 trimers were then further classified and aligned." However, there are no proper figures which

could elucidate the results properly? From where does the author determine 4% or 90%. From figures or results it is not clear. The author mentioned 3.2Å resolution, but no proper figures and readers have to search for the FSC curve.

The distribution of data is a major point and indeed deserves to be made more clear. Figure 1 now includes panel G, which clarifies the distribution in its entirety, and also compares this to that obtained by re-processing EMPIAR-10489 of native PDC from the same organism. The FSC curve is now also directly referenced in the text at the resolution claim, to figure S1.

3. The author should mark bad particles in his 2D class averages properly. Otherwise, delete that panel in figure S4. No need to show bad particle or 2D class averages. Most of the time, we do not show bad particles after 2D classifications.

The reviewer is correct. The bad 2D classes were omitted, since these indeed do not contribute significantly to the results and possibly confuse the reader.

4. Did the author try C1 symmetry? If the author applied C1 symmetry, does the model identical to this current model?

Indeed, reconstruction without enforced symmetry was attempted. Unfortunately, no significant asymmetry could be confidently observed. This is in all likelihood due to the flexibility of the CBD trimer, which constitutes variable asymmetry. This is corroborated by the molecular dynamics simulations undertaken since the first submission. The original manuscript did allude to this: *“Flexibility in the CBD is evident from data processing(Fig. S7).”* This has now been expanded: *“Since such flexibility results in variable asymmetry of the CBD trimer, no asymmetric features could be confidently established using cryo-EM data”*

5. How did the authors generate the initial model? It is important to show it properly.

The reviewer is entirely correct, this is an important part of rigorous cryo-EM workflows. To that end, the methods now clearly state that *“The ab-initio algorithm in RELION was used to initialize the optimization, in order to minimize reference bias(Zivanov et al, 2018)”*

6. Figure 1C, E3BP CBD (green) region of CTD:CBD sub-complex looks extremely low resolution. It will be better if the authors show the 3D model at a different orientation. This will help us to analyze the quality of the 3D model. Also, the author should fit the atomic model into the EM density map to show the overall agreement between the EM map and the atomic model.

Figure 1 now shows the consensus reconstruction in the same orientation as the improved reconstruction. This is now panel 1D. The atomic model has also been fitted into the density mesh to convey overall fit, in panel 1F.

7. The atomic model 7R5M builds based on the cryo-EM map E2-E3BP complex (EMDB 14331). However, several amino acids (Ser303-Arg310, Ala226-Val247, and so on) do not fit in the EM map at all. So how do authors build the atomic model of these regions when EM density maps are not visible at all. This is one major concern I have, and authors should rebuild the atomic model.

The atomic model was first built against the partial reconstruction following symmetry expansion, which was contained to a region small enough to limit the inherent flexibility and make the CBD trimer a significant portion to maximize reconstructed quality. As a result, E2 chains adjacent to the main E2 trimer were not completely included. The entirety of those chains were nonetheless built for completeness, such that parts of those chains were not supported by density. The reviewer justifiably raises the concern that this is not appropriate, as the atomic model should only extend to regions with support by the cryo-EM reconstruction. The atomic model has thus been reduced to those regions, in line with the reviewer's concern. The deposited pdb model has been revised and a new validation report generated. This report is attached to this response.

Authors may argue that they built an atomic model based on enhanced.mrc map. However, in case of enhanced.mrc, volume threshold abruptly ended. This is only possible when researchers use a very tight mask or build any mrc model from pdb structure.

The map post-processed by deepEMhancer does not display the conventional solvent background, due to the implementation details of the program. The abrupt end to the reconstruction histogram is thus *not* due to use of a mask or generation of a map from a molecular model. The latter would in fact constitute scientific misconduct, as it would imply that the density used to fit the atomic model would in fact be fabricated. The reviewer can be assured that this is not the case, and is encouraged to verify that the reconstruction histogram of the provided map is indeed expected, by using the program on a map of their own choosing. The lack of background noise in deepEMhancer output is also visible in Figures S10, S11, and S12 of Sanchez-garcia *et al*, (2021).

Additionally, atomic model 7R5M can fit into enhanced.mrc at 0.0337 threshold, however, several extra densities are visible near the EM core map, and the author did not try to explain it properly. I think the author should pay more attention to cryo-EM data processing and model building.

The extra densities that the reviewer observes are the core-internal loops of the CTD trimers from other parts of the PDC core assembly. This is now directly referenced in the text: *"The same loop of adjacent CTD trimers within the PDC core assembly can also be observed in the periphery of the localized reconstruction, near the center of the PDC core."*

8. These are attributed to false-positive identification within the margin of error of data classification. 90% of PDC core particles had at least one CBD-trimer bound. How do

authors determine that "the 90% of PDC core particles had at least one CBD-trimer bound"?

The reviewer is right to point out that this is not clear from the context. To rectify this, the distribution data is now included in panel G of Figure 1, and are explicitly mentioned in the text: *"CBD-occupied E2-trimers were selected by classification, and mapped back to the cores particles from which they were extracted, which permitted analysis of how many CBD trimers were found within each core particle (Fig 1G). The majority of E2 core particles were had 3-4 CBD trimers assigned, whereas only 4% were assigned 5 or more CBD-trimers. This agrees well with a limit of 4 due to steric considerations(Forsberg et al, 2020), and these are thus attributed to false positive identification within the margin of error of data classification."*

9. "One would expect F346 and possibly L385 to pack against I299 and F393,connected to the CBD by structured loops, and is responsible for binding to E2 (Fig. 2B)." From Figure 2B, none of these are clear. Better representation of figures is highly recommended.

Panels C and D of Figure 2 now make the packing much clearer. The structured loops are shown in panel C of Figure 2, with supporting evidence from molecular dynamics simulations as shown further in Figure S4.

10. "R301 is highly conserved across Pez fungi and stabilizes the C-terminal end of the same CBD monomer." Is there any biochemical evidence about R301 that it is stabilized C-terminal end? Did the author perform any mutation studies to validate the importance of residues L285, I289, V291, P420, L423, and V424 in the CBD helix? The author should perform a mutation study of Y342 to validate the interaction with E401 and R404. Additionally, the author should clearly mark M2 and M3 regions in the figure. It is extremely difficult to follow which is M2 and M3. Additionally, the authors predicted that K332 interacts with E409, K336 with T414, and so on. However, there is no biochemical evidence provided about these interactions. Is there any published evidence about these interactions and the importance of these amino acid residues?

The reviewer is correct in asking for corroborating evidence to the claims made. To that end, molecular dynamics simulations were carried out to validate the relevance of the indicated residues. That evidence will not be restated here, but is taken into account and valued accordingly in the main text. Phrasing is less speculative, referring directly to observed and quantified interactions. Interactions not supported by molecular dynamics have been omitted. We thank the reviewer for prompting this analysis, which has much improved the manuscript.

11. Authors should pay more attention to writing the methods like Model Building, Data Processing, Computational Structure Prediction, Phylogenetic analysis, and so on.

All these sections have been revised, and frequently expanded. Further details can of course be provided if more specific feedback is provided.

12. Figure S4 legend is very confusing. Like -Sym flag and IcosaCompatWithDefaultTetra.Sym? Why it is important of this type of figure legend. Generally, we don't write all these details in figure legends.

The information pertaining to methods that was included in the figure caption has been moved to supplementary methods, and includes more a complete clarification of terms that the general reader might not be familiar with.

Reviewer #2 (Remarks to the Author):

In this article, the Author presents the 3.2 Å resolution cryo-EM structure of the fungal E3-binding protein (E3BP) in complex with the PDC core, both from *N. crassa*. The measured structural data is complemented by modeling of the PSBD domain and its interactions with the PDC E1 and E3 proteins. Another finding presented in the paper is related to the homology and evolution of eukaryotic and fungal E3BP-like proteins, for which the Author concluded that these are likely orthologs.

1. The figures are well formatted in general and the only recommendation from my part is for Figure 1. In this figure, the domain topology is presented for both E2 and E3BP, but it would help the readers if amino acid numbering would also be included.

Amino acid numbering has been included in Figure panel 1A.

2. Also a mention that the PSBD here was only modeled should be included.

The section concerning PSBD specificity no lead with "*The PSBD which anchors peripheral PDC components was omitted from the present cryo-EM sample, and was consequently not reconstructed or directly visualized*"

3. The description of the methods used is suitable for the topic, but for those not familiar with all the expression systems, a mention that the Rosetta bacterial strain is actually *E. coli* might be welcome.

This is now mentioned explicitly

I find the presented work to be novel and suitable for publication. It also contains important conclusions for this field.

Reviewer #3 (Remarks to the Author):

The article entitled “The structure and evolutionary diversity of the fungal E3-binding protein” by Forsberg reports the structure of the *N. crassa* recombinantly expressed C-terminal catalytic transacetylase domain of E2 (CTD) and C-terminal core binding domain of E3BP (CBD) domains utilizing cryoEM at gold-standard 3.2 Å resolution. Based on that and on a broad sequence analysis the author attempts an orthology analysis comparing fungal and mammalian E3BP. Novel findings within the same context are introduced here by the author as the conservation and specificity of the PSBD of E3BP for E3 is also analyzed although it was not part of the structural study. Finally, the author runs alphafold to predict binding regions for the E3.

As an overall impression, the manuscript is a continuation of the previously published wonderful manuscript of the author (Forsberg et al, Nat Com 2020) that allows more accurate localization of E3BP domain arrangement in respect to the E2 core domains of PDC and a focus on the interacting residues. This submitted manuscript is a purely computational work where data of the previous paper were re-analyzed. The validity of the rest of the data presented here though relies mostly on sequence alignments and predictions without adding experimental verification. Especially the PSBD binding motifs computational prediction could have been validated by an experimental system. Nevertheless, the data presented could enrich current knowledge in PDC and are worth being communicated. The author might find useful the following major and minor points:

Major points

1. Parts of Figure 1 are already shown in the author's previous nature comm paper so I suggest to cite it. The Nature comm paper also included the PDC from *N. crassa* purified, and the images are available in the EMPIAR by the author. It would be perfectly matching to this manuscript if the claims from the coexpressed E2/E3BP (which were also performed for yeast in 1997) are substantiated in this manuscript as well. This means that image analysis and cryo-EM reconstructions along with 3D models of E2 and E3BP are retrieved from *N. crassa* PDC using his current cryo-EM reconstruction as template and compare to further understand key points raised in the manuscript. Even if achieved resolution would be lower, stoichiometries, potential interfaces etc would add a lot to the current draft.

EMPIAR 10489 has now been analyzed in the same manner, and the data is represented more transparently in Figure panel 1G. The resolution is predictably worse (as the reviewer also expected), and the ability to distinguish CBD binding appears slightly worse (presumably due to the presence of peripheral components that increase overall noise). The overall results hold well and agree with the native PDC distribution, especially when grouped by the most likely number of CBD monomers assuming that they all participate in trimers (Figure panel 1G, inset.)

2. Figure 2 - Lack of PDB validation – depositing more structures from cryo-EM maps calculated here are required. The author does not provide the EMDB/PDB validation report along with the exact deposited PDB model/ EMDB cryo-EM map to assess the quality of the model and its fit in the density – only a table is provided, with e.g. clashscore of low quality (must be better than 2). Currently, due to the lack of these exact data, I cannot comment on the presented data related to cryo-EM. If work from major point (1) is done, then the equivalent validations and models should be provided in the next round of revisions.

The clashscore has been improved in combination with re-building and re-refinement, performed in response to concerns expressed by reviewer number 1. Supplementary table 1 has been updated with new refinement statistics. The final validation report of this revised model is also provided.

3. Figures 3 and 4. Topology analysis of E3BP CBD is not properly citing available literature. Literature related to the E3BP is currently limited in the submitted manuscript and its similarity to E2 is, e.g., known for decades – sequence comparisons and domain comparisons are indeed known. Therefore, the author should better review the PDC literature comparing E2 and E3BP to substantiate or down-tone novelty claims currently present in the manuscript.

The reviewer is correct in pointing out that the literature was insufficiently reviewed. Any claims to novelty have been revised and the following references have been added:

- a. Behal1989 (homology of E2 and E3BP within fungi)
- b. Lawson1991 (evidence of E3-binding of E3BP (then PX) in fungi)

If the reviewer feels that specific research articles should be included, any suggestions would of course be considered.

4. Comparisons of CBDs. The results of the currently interesting computational analysis of E2/E3BP PSBDs are already discussed in e.g. the Patel book 1996. Even mutational studies have been performed probing bindings of E1 and E3, and even works related to the linker regions and their lengths have thoroughly investigated this. It is suggested that either (some) experimental validation of (some) AlphaFold-based computational models is performed, or the author moves these data to the discussion section with proper referencing.

By “the Patel book 1996”, does the reviewer refer to doi:10.1007/978-3-0348-8981-0?

The reviewer justifiably desires that the present analysis of the PSBD specificity should be placed in the context of existing work. Indeed, mutational studies are already cited (Cizak2006, Brautigam2006), and critical residues identified in those studies are discussed (e.g. human E3BP I157 identified in Cizak2006). This section has been reformatted to make this more clear.

The reason for conducting this analysis has also been clarified as part of this re-formatting: the aim of analyzing the PSBD is to validate the orthology of E3BP and E2 in eukaryotes and provide distinguishing features that may suggest avenues for future inquiry into the specificity for E2 or E3. As part of this, the predicted models have also been down-played, now clearly presented only as a possible explanation of the observed sequence conservation C-terminal to the canonical PSBD in Ascomycota.:

*“**Predicted** models of the binding of the E3BP PSBD to E3 indicate the termination motif may result in different conformations comparing human and *N.crassa*. Whereas human I228 is **predicted** to fold back onto the PSBD itself (Fig. S9A), *N.crassa* I214 instead approaches the previously mentioned and highly conserved L208.”*

...

*“To examine if this motif offers a further rationale to E3 specificity, the *N.crassa* E3-PSBD complex predicted by alphafold was considered (Fig. S9B). In this **prediction**, ...”*

Finally, figure 5 has been made a supplementary figure and merged with what was S5 (now S9), to emphasize that this is used to aid analysis but is not a main finding presented as results.

5. Currently, the manuscript includes experimental data only for the E3BP trimer which were previously collected and included in the Nat Comm 2020 paper. If this is not the case, of course detailed methods and supp should be provided with validations for these new constructs, preparations etc.

The reviewer is correct in that no new data was collected. The procedures used to prepare samples and collect the data are nevertheless presented for completeness and posterity.

6. Please correct typos and inconsistencies in the manuscript (some are mentioned in the minor points) because it was difficult for me to keep track.

Apologies for any spelling mistakes. These should now be entirely rectified.

Minor points

7. Line 14-15: author mentions that “E3 retention is mediated by the E3-binding protein (E3BP), which has not previously been clearly resolved within the PDC.” which is not true as it has been previously shown by the author himself, in Kyrilis et al Cell reports 2021 (incorrectly cited), in Tüting et al, and later on in Skalidis et al (not cited here). Proteomic, activity, western blot, and cryo-EM reconstruction data have supported this retention.

What was meant here is that it has not been resolved to this degree by direct reconstruction, but the reviewer's criticism is warranted. This has been rephrased as “*which is here resolved within the PDC core from *N.crassa*, resolved to 3.2Å*”

8. Please revise also claims in Line 45.

In the submitted manuscript, line 45 reads “*Unlike mammals however, fungal E3BP binds to the interior of the E2 core assembly instead of substituting core components(Forsberg et al, 2020; Skalidis et al, 2021; Tüting et al, 2021)(Fig. 1B).*” It is not clear in what way the reviewer disagrees with this, when the preceding critique was that this has long been established. Any claim that the reviewer feels to be exaggerated can be subject to change of course, but this critique should perhaps be clarified.

9. Line 59: the word “either” maybe should be removed or sentence rephrased?

Duly amended

10. Line 73: typo of word “identification”

Duly amended

11. Line 83: residues P420 and L423 are not properly annotated in Figure 2D. maybe the author could change the illustration to show the full interface? or mention that these residues are not shown. The author can also show the complete CBD and its fit in the high-resolution density in a systematic manner, also by rotating the illustration to show the flanking beta strands as well.

Figure 2 has been greatly revised, and these residues are now shown in panel 2F.

12. Line 94: I think numbering is wrong as it should be 347-389 and not 289, please check.

Indeed, it has been duly amended

13. Line 100: the author here stretches a bit the data available as he claims that ONLY M2 is the domain that binds the E2 but the M3 is not clearly resolved thus cannot be excluded as an interacting interface, it would be nice to comment on this, especially when the disordered linker is not included.

The final section of the leading paragraph does allude to this, which has been clarified to read “*M3 might thus be a secondary binding motif. Residual density is observed in the CTD binding pockets where trimeric CBD is sterically impermissible (Fig. S6B), but as it cannot be clearly resolved auxiliary M3 binding should be considered possible but unsubstantiated. Conducted simulations also did not indicate a preference for the M3 region to approach or favorably interact with available binding pockets. The observed density could alternatively be M2 of meta-stable monomeric CBD, that does participate CBD trimers. The universal conservation of M3 thus remains unexplained*”

14. Line 101: typo“an” amphipathic helix... instead of “a”

Duly amended

15. Lines 109-111: very hypothetical in my opinion, please down-tone the claims.

In the submitted manuscript, line 109-111 reads “*One must consider that M3 might act as a secondary binding motif against the CTD binding pocket. Residual density is observed in these interfaces (Fig. S6B), but cannot be clearly resolved. This density could alternatively be meta-stably bound M2 of monomeric CBD that does not form CBD trimers.*” The reviewer pointed out in minor point 13 that M3 might be such a binding motif. Nonetheless, the statement now affirms that “*auxiliary M3 binding should thus be considered possible but unsubstantiated*” in lieu of structured density to support such a notion. The alternative

16. Line 119: Residues K403_CBD-D412-CTD are wrong according to the figure- maybe the author means R425_CBD-D412-CTD?

The additional evidence provided by molecular dynamics does not support these interactions, and they are consequently not mentioned in the main text any longer.

17. Line 125: the author cites Tüting et al, 2021 manuscript here but does not include it in the comparison of eg Figure S6, nor the model organism.

By their own account, Tüting et al constructed an atomic model of the CBD including sidechains, based on a 6.2Å map. They refined this using Phenix, but only provided the PDB as an auxiliary file in an SBgrid database entry. No refinement statistics were provided. While a comparison could be made, the lack of scientific merit in the Tüting model makes such a comparison potentially misleading. Differences that might be attributed to differences between species might simply be due to poor refinement procedure or model statistics. Hence, a direct comparison is not advisable. The model organism used by Tüting et al has however been annotated in Figure S6 (now S8) to place it in the phylogenetic context presented here.

18. Line 144: words “unambiguously” and “throughout” have typos

Duly amended

19. Line 148-149: sentence that starts with “Were...” I think should be rephrased, maybe it is “Where...”?

Duly amended to “*If core-substitution would be possible*”

20. Line 151-152: author mentions that there are some search criteria in Figure S6 that I don't see. These criteria are not clear to me. Fig S6 shows phylogenetic relationship of fungal species

This sentence indicates that the criteria has resulted in the tripartite division, which is shown in fig S6. The phrasing has been changed to clarify this.

21. Figure S2 is a bit difficult for me to understand, there are typos in the legend, and in general can you please optimize it? The distances that are presented are not described in the legend.

Figure S2 (now S6) has been duly amended. The Figure is more clear with colors corresponding to other figures. Distances are mentioned explicitly and the wording has been made more clear.

22. Line 152-154: I think the author here stretches out a bit the available data to make this conclusion, and also reports the obvious of the eukaryotic origin of E3BP.

The shared evolutionary origin of E2 and E3BP has been established prior to the present paper, and any phrasing that might suggest otherwise has been changed to avoid confusion. However, the joint evolutionary origin of E3BP across eukaryotic species has not been established, and is not at all certain from existing literature and evidence. The present work however provides evidence to support this notion. The sentence has however been changed to the following, to assure claims are not exaggerated: *“It therefore seems likely that fungal and animal E3BP are indeed orthologs, with a shared evolutionary origin, and that E3BP is a eukaryotic gene”*.

23. Line 160: word “analyzed” has a typo

Duly amended

24. Line 197: word “specific” typo

Duly amended

25. Line 198: words “with this” mistyped

Duly amended

26. Line 237: is it Fig. S6 or it should be Fig. S7? Please double-check citations to Figures and Tables of the manuscript because I might have missed something.

Duly amended

27. Line 245: “....for the both the...” rephrase

Duly amended

28. Line 263: “Pez M3 is however more similar to Zyg M3 that to its own M2....” Is there a typo here?

Duly amended

29. Line 269: word “Within” has a typo

Duly amended

30. Line 298: ...the PSBD...” space missing

Duly amended

31. Line 307: typo instead of “to”, “co is written

Duly amended

32. Line 315 and 324: Celsius miss the degree symbol?

Duly amended

33. Line 376: “s” missing from word “protein” as should be plural

Duly amended

34. Line 520: delete word “the” after “...hydrophobicity that stabilizes...”

Duly amended

35. Line 530: "...symmetry axis" There is a typo here and further on the same line it is either "additional" or "addition of"

Duly amended

36. Line 531: "from utilizing ny of the remaining 3 binding sites that are most proximal to both G347 and V392." Typo

Duly amended

37. Line 532: typo of word "either"

Duly amended

38. In general in the text and also in Supp. Fig S2 the author claims these residual density in the binding pocket. If it was E3BP then also more densities of the E3BP would be visible????maybe the author could explain if he saw more than this except the disordered parts

The densities are not claimed, they are simply observed. Their origin is discussed in the context, and attributed either to M3 or M2 of monomeric CBDs which do not participate in canonical CBD trimers. Due to the symmetric binding interface, monomeric CBD would present the challenge of a mixture of binding orientations in addition to flexibility, inherent disorder and a small structured domain. The very limited flexibility of the CBD trimer was just enough to be classifiable in the present data, indicating that the increased challenge of extending these residual densities by further analysis is unlikely to be successful. Biochemical approaches would be needed to more confidently attribute these densities to either M2 or M3.

39. Line 573: A, B and C are missing from the Fig. S5 even if mentioned in the legend

This has been amended, and S5 is now S8

40. Line 577: word "if" is a typo

Duly amended

41. Supp Fig S4, please put scale bar in the cryo-EM image

Added, and the figure re-organized as feedback from per reviewer 2.

Reviewers' comments:

Reviewer #3 (Remarks to the Author):

The author has substantially revised and improved his already interesting manuscript on the structure of the coexpressed E2/E3BP construct. The resolution achieved for this target is astounding. This reviewer is not an expert in MD simulations, and, therefore, cannot comment on the newly revised part of the manuscript but most minor and major points this reviewer raised have been answered. Few remaining minor comments are appended below:

Minor comments to previous major points:

- 1) This reviewer thanks the author for the additional analysis performed on the dataset of EMPIAR 10489, it is really insightful. It would be useful if this analysis was open and accessible, e.g., reconstructions, class averages, mrc files, fsc plots etc.
- 2) Indeed, the author has provided very much improved statistics. However, the author can also comment on major point 1 and availability of the analysis of the endogenous data.
- 3) The updated reference list is improved. The author should also keep in mind that Maeng et al 1994 showed the 12 E3BPs in yeast with AUC.
- 4) Yes, this is the book this reviewer referenced and the contextualization is appreciated.
- 5) If this is a reanalysis of existing data, the author should clarify this in their manuscript, then.
- 6) The paper has very minor inconsistencies now, which I guess will be addressed at the proofreading stage.

Lastly, the author may also reflect on the challenges in resolving an endogenous molecule like PDC. Analyzing a pure, overexpressed, possibly truncated, construct (however difficult it may be) is an approximation for explaining in-cell function. The author has himself performed endogenous PDC analysis and observed that resolution of the E3BP remained low. But this does not mean that the structure cannot be modeled, even at lower confidence: A model is just a model, and of course, defined by resolution as well as supporting biochemical and biophysical data. For minor comment 8, this reviewer means (a) reference is not Skolidis et al 2021 (no such paper exists in 2021 but is in 2022), but Kyrilis et al 2021. In addition, the Stoops et al 1997 yeast paper should be cited here, which was the first one to observe the E3BP in the interior of the core, mentioning "It was surprising then to find in the case of *S. cerevisiae* PDC, that major portions of tBP, BP, and BP·E3 lie inside the tE2 core."

Overall, this reviewer would like to congratulate the author for his astounding work on the E3BP. This paper will become a benchmark on E3BP, and overall, its contribution to understanding PDC.

Reviewer #4 (Remarks to the Author):

I was quite excited by the idea to see this work on the (once upon a time) so-called protein X, a mysterious component of the pyruvate dehydrogenase complex. Disappointingly, I have to frankly indicate that my enthusiasm rapidly vanished because I found the manuscript very difficult to read. The description of the structure starts with the following sentence "Tetrahedral symmetry was used for the preliminary reconstruction". The pyruvate dehydrogenase core is not tetrahedral but icosahedral. So, there is a symmetry mismatch. This issue is not addressed or explained at all. Then the manuscript states that "The CBD of *N. crassa* has a trimeric interface with a symmetry axis that coincides with that of the core E2 CTD-trimer it binds to (Fig. 2A). This trimeric interface has evolved from the dimeric interface that core CTD trimers form to form larger assemblies." From the first sentence, one would gather that the CBD trimer should associate to the CTD trimer. However, Figure 2A shows that the CBD trimer associates to subunits belonging to two different CTD trimers. This feature is probably implied in the second sentence of the above text. However, this text remains

obscure to me. How can a trimeric interface evolve from a dimeric interface? The dimer interface in the pyruvate dehydrogenase is established by the C-terminal residues of the catalytic domain whereas the trimeric interface of the CBD does not seem to involve the C-terminal residues. And how can a three-fold symmetric trimeric protein interact with two subunits belonging to a trimer and one belonging to a second two-fold related trimer? Again, there is no description of this feature. Figure 2 is the critical figure to assess this point. However, my impression is that there is a mismatch between the panels and figures legends. Panel B is described as "The binding motif is centered around a hydrophobic patch on the CTD complementary to CBD F324, and charge complementarity to CTD lysines K266 and K263". Yet, I cannot see F324, K266 and K263 in the panel. My feeling is that this legend might apply to panel E but I am not sure. Here, I became confused. Reading the manuscript further did not really help to clarify and understand the reported data. It is a pity because they are potentially very interesting. I think that this manuscript should be re-written. I also have the impression that the comments by the previous reviewers about the cryoEM procedures are not fully addressed.

Reviewer #5 (Remarks to the Author):

Overall the author responses to the reviewer 1 are logical. The method section for cryoEM data analysis is clear in the revised manuscript. A few minor suggestions:

1- Show local resolution estimation by coloring the cryoEM density map in Figure 1E.

2- Regarding figure 2, indicate the distances shown by the dash lines. Also, regarding residues involved in salt bridges, are they supported by cryoEM density? Same goes for the structured loops. Perhaps a supplementary figure can be used to show these densities.

3- The author can consider performing evolutionary coupling analysis (e.g., PMID: 30304492, server: <https://evcouplings.org/>) to further support interactions between Y342 with E401 and R404 residues or any other predicted residue-residue interactions.

4- Figure S4, label the Y-axis

Response to referees (2nd)

Structure and evolutionary diversity of the fungal E3-binding protein

Bjoern O. Forsberg

COMMSBIO-22-1397-B

2023-03-12

Reviewer comments are in **black**. They have been numbered for easy referencing.

Response is shown in **blue**.

Reviewer #3 (Remarks to the Author):

The author has substantially revised and improved his already interesting manuscript on the structure of the coexpressed E2/E3BP construct. The resolution achieved for this target is astounding. This reviewer is not an expert in MD simulations, and, therefore, cannot comment on the newly revised part of the manuscript but most minor and major points this reviewer raised have been answered.

Few remaining minor comments are appended below:

Minor comments to previous major points:

1. This reviewer thanks the author for the additional analysis performed on the dataset of EMPIAR 10489, it is really insightful. It would be useful if this analysis was open and accessible, e.g., reconstructions, class averages, mrc files, fsc plots etc.

The reviewer is correct. A new reconstruction EMD-16884 and an associated PDB model 8OHS has been deposited, depicting the reconstruction from positively classified CBD-trimers in the native PDC data (EMPIAR-10489). In addition, the processing pipeline and associated data is depicted in a new supplementary figure. The methods section also provides processing details.

2. Indeed, the author has provided very much improved statistics. However, the author can also comment on major point 1 and availability of the analysis of the endogenous data.

This validation analysis is now included.

3. The updated reference list is improved. The author should also keep in mind that Maeng et al 1994 showed the 12 E3BPs in yeast with AUC.

This is true, and is now remarked in the text when a comparison is made with structural implications of differences between ascomycota and saccaromyces E3BP: *"In the absence of such steric restraints, additive affinity may similarly enforce a limit of 15 bound copies of E3BP"*

in Sac. Of note, Sac PDC has been previously observed to contain 12 E3BP(Maeng et al, 1994), which instead implies that this too provides steric occlusion by trimerisation.”

4. Yes, this is the book this reviewer referenced and the contextualization is appreciated.
5. If this is a reanalysis of existing data, the author should clarify this in their manuscript, then.

A note to this effect is now made under the methods heading “Sample preparation and data collection”, which now also references details provided in the supplementary methods section.

6. The paper has very minor inconsistencies now, which I guess will be addressed at the proofreading stage.

Lastly, the author may also reflect on the challenges in resolving an endogenous molecule like PDC. Analyzing a pure, overexpressed, possibly truncated, construct (however difficult it may be) is an approximation for explaining in-cell function. The author has himself performed endogenous PDC analysis and observed that resolution of the E3BP remained low. But this does not mean that the structure cannot be modeled, even at lower confidence: A model is just a model, and of course, defined by resolution as well as supporting biochemical and biophysical data.

The resolution obtained for the reprocessed native PDC (EMD-16884, 4.1Å) only supports rigid-body fitting of individual chains from model (7r5m) established using the reconstruction of the reconstituted subcomplex (EMD-14331, 3.3Å). The limited resolution precludes a confident analysis of differences in the CBD between the two, and the only major finding that is supported by the data is to overall corroborate the existence and predominance of the trimeric form of the CBD. This is now also noted in the text: *“The analysis was repeated using native PDC data (EMPIAR-10489), which found a highly similar CBD-trimer distribution. Increased uncertainty in both classified proportions and attained were however noted due to the higher overall noise in this data, to the extent that it can only quantifiably corroborate the existence and predominance of the CBD trimer in native samples compared to the recombinant expression, and conversely cannot confirm specific interactions observed in the latter.”*

For minor comment 8, this reviewer means (a) reference is not Skalidis et al 2021 (no such paper exists in 2021 but is in 2022), but Kyrilis et al 2021. In addition, the Stoops et al 1997 yeast paper should be cited here, which was the first one to observe the E3BP in the interior of the core, mentioning “It was surprising then to find in the case of *S. cerevisiae* PDC, that major portions of tBP, BP, and BP·E3 lie inside the tE2 core.”

The misquoted reference has been duly changed and a note to the effect that Stoops et al observed PDC core-interior density has been made.

Overall, this reviewer would like to congratulate the author for his astounding work on the E3BP. This paper will become a benchmark on E3BP, and overall, its contribution to understanding PDC.

Reviewer #4

I was quite excited by the idea to see this work on the (once upon a time) so-called protein X, a mysterious component of the pyruvate dehydrogenase complex. Disappointingly, I have to frankly indicate that my enthusiasm rapidly vanished because I found the manuscript very difficult to read.

The description of the structure starts with the following sentence "Tetrahedral symmetry was used for the preliminary reconstruction". The pyruvate dehydrogenase core is not tetrahedral but icosahedral. So, there is a symmetry mismatch. This issue is not addressed or explained at all.

The methods section presently contains an adequate clarification of how processing was conducted and its motivation. To further elaborate on the reviewer's specific concern, tetrahedral symmetry was used since the intention was to reveal the characteristic of a protein on the 3-fold symmetry axis of the PDC. It was thus easier to perform particle centering and fundamental alignments in a symmetry where the 3-fold symmetry was aligned with the z-axis. No such icosahedral symmetry convention is commonly available and implemented in cryo-EM refinement programs, thus tetrahedral symmetry was applied. This is not an issue, since symmetry-expansion of fundamental alignment of the PDC core is not dependent on the preliminary symmetry, and since tetrahedral symmetry is a sub-symmetry of icosahedral symmetry where each 5-fold symmetry axis is broken. The symmetry mismatch thus simply omits a 5-fold symmetry averaging that is non-essential for the present processing. Moreover, the E3BP aligns tetrahedrally interior to the core at maximal occupancy, as shown previously.

Then the manuscript states that "The CBD of *N.crassa* has a trimeric interface with a symmetry axis that coincides with that of the core E2 CTD-trimer it binds to (Fig. 2A). This trimeric interface has evolved from the dimeric interface that core CTD trimers form to form larger assemblies." From the first sentence, one would gather that the CBD trimer should associate to the CTD trimer. However, Figure 2A shows that the CBD trimer associates to subunits belonging to two different CTD trimers. This feature is probably implied in the second sentence of the above text.

The first sentence has been amended to "*The CBD of N.crassa has a trimeric interface with a symmetry axis that coincides with that of a core E2 CTD-trimer (Fig. 2A).*" as to not indicate direct association. The second sentence was also amended to directly reference the CBD trimer interface as separate from how the CBD associates with the core (this is the topic of a dedicated

section in the results): *“The CBD trimer interface has evolved from the dimeric interface that core CTD trimers form to extend to larger assemblies”*

However, this text remains obscure to me. How can a trimeric interface evolve from a dimeric interface? The dimer interface in the pyruvate dehydrogenase is established by the C-terminal residues of the catalytic domain whereas the trimeric interface of the CBD does not seem to involve the C-terminal residues.

The trimeric CBD interface does involve its C-terminus, as is evident from the deposited model, mentioned in the text (“...stabilizing the CBD C-terminus”), and majorly part of the MD analysis (shown explicitly in figure S4C+D). It has also been clarified at its first mention: *“It is hydrophobic in nature, formed largely from the C-terminal end of the first CBD helix (residues L285, I289, and V291), as well as its C-terminus(Fig. 2B)”*

And how can a three-fold symmetric trimeric protein interact with two subunits belonging to a trimer and one belonging to a second two-fold related trimer? Again, there is no description of this feature.

The reviewer has inferred associations between the PDC core and the CBD trimer that are not evident in the reconstruction, model, or any figure. The text has been amended to avoid such interpretations, and the present formulation provides an accurate depiction of the spatial relationship between the CBD trimer and the core together with figures 1C+F, 2A, S3A+B, and S6, which thus fully clarify the present structure.

Figure 2 is the critical figure to assess this point. However, my impression is that there is a mismatch between the panels and figure legends. Panel B is described as “The binding motif is centered around a hydrophobic patch on the CTD complementary to CBD F324, and charge complementarity to CTD lysines K266 and K263”. Yet, I cannot see F324, K266 and K263 in the panel. My feeling is that this legend might apply to panel E but I am not sure. Here, I became confused.

The reviewer correctly observes that the figure legend labels are mismatched. This has been amended.

Reading the manuscript further did not really help to clarify and understand the reported data. It is a pity because they are potentially very interesting. I think that this manuscript should be re-written. I also have the impression that the comments by the previous reviewers about the cryoEM procedures are not fully addressed.

The opinions expressed here by the reviewer are unfortunately not constructive enough to merit changes to the manuscript.

Reviewer #5

Overall the author responses to the reviewer 1 are logical. The method section for cryoEM data analysis is clear in the revised manuscript. A few minor suggestions:

1. Show local resolution estimation by coloring the cryoEM density map in Figure 1E.

This has been implemented.

2. Regarding figure 2, indicate the distances shown by the dash lines. Also, regarding residues involved in salt bridges, are they supported by cryoEM density? Same goes for the structured loops. Perhaps a supplementary figure can be used to show these densities.

Distances have been added to any dashed lines in Figure 2. A note has also been made in the figure caption to the effect that these distances are instantaneous measurements of what appears to be transient and variable but nonetheless persistently occurring interaction during MD simulations, and that the modeled distance is not necessarily inversely proportional to significance or strength of the interaction: "*Annotated distances are instantaneous distances of the deposited model PDB-7r5m, and not equilibrium or binding distances observed in simulations.*". This is also why cryo-EM density provides little support for static interactions in a way that can be visualized by cryo-EM to corroborate the interaction.

3. The author can consider performing evolutionary coupling analysis (e.g., PMID: 30304492, server: <https://evcouplings.org/>) to further support interactions between Y342 with E401 and R404 residues or any other predicted residue-residue interactions.

The reviewer suggests a pertinent analysis that was considered previously. Unfortunately, no such pipeline has been found that is able to accurately and confidently assign pairwise correlations that yield meaningful interpretation for Fungal E2 and E3BP. Prompted by the reviewer, EVcouplings was nevertheless attempted, using *N. Crassa* sequences P20285 and Q7RWS2 as input for a "complex" prediction, using a Bit-score cutoff = 0.1. However, by visualizing the confident couplings (>0.9) it appears that the detected couplings map the homologous regions between the two proteins, rather than those that are in close proximity under complex formation. Without a fundamental understanding of the methods employed by EVcoupling, it can be speculated that the homology of E2 and E3BP thus constitutes a complicating factor in this analysis, which would obscure the intended correlative analysis. A faithful such analysis might be very informative, but thus demands efforts that are not in proportion to the gain compared to the present analysis in the given structural context.

4. Figure S4, label the Y-axis

Duly added

Below follow any issues and their responses, raised in previous rounds of review, for the sake of completeness.

Reviewer #2 (Remarks to the Author):

In this article, the Author presents the 3.2 Å resolution cryo-EM structure of the fungal E3-binding protein (E3BP) in complex with the PDC core, both from *N. crassa*. The measured structural data is complemented by modeling of the PSBD domain and its interactions with the PDC E1 and E3 proteins. Another finding presented in the paper is related to the homology and evolution of eukaryotic and fungal E3BP-like proteins, for which the Author concluded that these are likely orthologs.

1. The figures are well formatted in general and the only recommendation from my part is for Figure 1. In this figure, the domain topology is presented for both E2 and E3BP, but it would help the readers if amino acid numbering would also be included.

Amino acid numbering has been included in Figure panel 1A.

2. Also a mention that the PSBD here was only modeled should be included.

The section concerning PSBD specificity no lead with "*The PSBD which anchors peripheral PDC components was omitted from the present cryo-EM sample, and was consequently not reconstructed or directly visualized*"

3. The description of the methods used is suitable for the topic, but for those not familiar with all the expression systems, a mention that the Rosetta bacterial strain is actually *E. coli* might be welcome.

This is now mentioned explicitly

I find the presented work to be novel and suitable for publication. It also contains important conclusions for this field.

Reviewer #3 (Remarks to the Author):

The article entitled “The structure and evolutionary diversity of the fungal E3-binding protein” by Forsberg reports the structure of the *N. crassa* recombinantly expressed C-terminal catalytic transacetylase domain of E2 (CTD) and C-terminal core binding domain of E3BP (CBD) domains utilizing cryoEM at gold-standard 3.2 Å resolution. Based on that and on a broad sequence analysis the author attempts an orthology analysis comparing fungal and mammalian E3BP. Novel findings within the same context are introduced here by the author as the conservation and specificity of the PSBD of E3BP for E3 is also analyzed although it was not part of the structural study. Finally, the author runs alphafold to predict binding regions for the E3.

As an overall impression, the manuscript is a continuation of the previously published wonderful manuscript of the author (Forsberg et al, Nat Com 2020) that allows more accurate localization of E3BP domain arrangement in respect to the E2 core domains of PDC and a focus on the interacting residues. This submitted manuscript is a purely computational work where data of the previous paper were re-analyzed. The validity of the rest of the data presented here though relies mostly on sequence alignments and predictions without adding experimental verification. Especially the PSBD binding motifs computational prediction could have been validated by an experimental system. Nevertheless, the data presented could enrich current knowledge in PDC and are worth being communicated. The author might find useful the following major and minor points:

Major points

1. Parts of Figure 1 are already shown in the author's previous nature comm paper so I suggest to cite it. The Nature comm paper also included the PDC from *N. crassa* purified, and the images are available in the EMPIAR by the author. It would be perfectly matching to this manuscript if the claims from the coexpressed E2/E3BP (which were also performed for yeast in 1997) are substantiated in this manuscript as well. This means that image analysis and cryo-EM reconstructions along with 3D models of E2 and E3BP are retrieved from *N. crassa* PDC using his current cryo-EM reconstruction as template and compare to further understand key points raised in the manuscript. Even if achieved resolution would be lower, stoichiometries, potential interfaces etc would add a lot to the current draft.

EMPIAR 10489 has now been analyzed in the same manner, and the data is represented more transparently in Figure panel 1G. The resolution is predictably worse (as the reviewer also expected), and the ability to distinguish CBD binding appears slightly worse (presumably due to the presence of peripheral components that increase overall noise). The overall results hold well and agree with the native PDC distribution, especially when grouped by the most likely number of CBD monomers assuming that they all participate in trimers (Figure panel 1G, inset.)

2. Figure 2 - Lack of PDB validation – depositing more structures from cryo-EM maps calculated here are required. The author does not provide the EMDB/PDB validation report along with the exact deposited PDB model/ EMDB cryo-EM map to assess the quality of the model and its fit in the density – only a table is provided, with e.g. clashscore of low quality (must be better than 2). Currently, due to the lack of these exact data, I cannot comment on the presented data related to cryo-EM. If work from major point (1) is done, then the equivalent validations and models should be provided in the next round of revisions.

The clashscore has been improved in combination with re-building and re-refinement, performed in response to concerns expressed by reviewer number 1. Supplementary table 1 has been updated with new refinement statistics. The final validation report of this revised model is also provided.

3. Figures 3 and 4. Topology analysis of E3BP CBD is not properly citing available literature. Literature related to the E3BP is currently limited in the submitted manuscript and its similarity to E2 is, e.g., known for decades – sequence comparisons and domain comparisons are indeed known. Therefore, the author should better review the PDC literature comparing E2 and E3BP to substantiate or down-tone novelty claims currently present in the manuscript.

The reviewer is correct in pointing out that the literature was insufficiently reviewed. Any claims to novelty have been revised and the following references have been added:

- a. Behal1989 (homology of E2 and E3BP within fungi)
- b. Lawson1991 (evidence of E3-binding of E3BP (then PX) in fungi)

If the reviewer feels that specific research articles should be included, any suggestions would of course be considered.

4. Comparisons of CBDs. The results of the currently interesting computational analysis of E2/E3BP PSBDs are already discussed in e.g. the Patel book 1996. Even mutational studies have been performed probing bindings of E1 and E3, and even works related to the linker regions and their lengths have thoroughly investigated this. It is suggested that either (some) experimental validation of (some) AlphaFold-based computational models is performed, or the author moves these data to the discussion section with proper referencing.

By “the Patel book 1996”, does the reviewer refer to doi:10.1007/978-3-0348-8981-0?

The reviewer justifiably desires that the present analysis of the PSBD specificity should be placed in the context of existing work. Indeed, mutational studies are already cited (Cizak2006, Brautigam2006), and critical residues identified in those studies are discussed (e.g. human E3BP I157 identified in Cizak2006). This section has been reformatted to make this more clear.

The reason for conducting this analysis has also been clarified as part of this re-formatting: the aim of analyzing the PSBD is to validate the orthology of E3BP and E2 in eukaryotes and provide distinguishing features that may suggest avenues for future inquiry into the specificity for E2 or E3. As part of this, the predicted models have also been down-played, now clearly presented only as a possible explanation of the observed sequence conservation C-terminal to the canonical PSBD in Ascomycota.:

*“**Predicted** models of the binding of the E3BP PSBD to E3 indicate the termination motif may result in different conformations comparing human and *N.crassa*. Whereas human I228 is **predicted** to fold back onto the PSBD itself (Fig. S9A), *N.crassa* I214 instead approaches the previously mentioned and highly conserved L208.”*

...

*“To examine if this motif offers a further rationale to E3 specificity, the *N.crassa* E3-PSBD complex predicted by alphafold was considered (Fig. S9B). In this **prediction**, ...”*

Finally, figure 5 has been made a supplementary figure and merged with what was S5 (now S9), to emphasize that this is used to aid analysis but is not a main finding presented as results.

5. Currently, the manuscript includes experimental data only for the E3BP trimer which were previously collected and included in the Nat Comm 2020 paper. If this is not the case, of course detailed methods and supp should be provided with validations for these new constructs, preparations etc.

The reviewer is correct in that no new data was collected. The procedures used to prepare samples and collect the data are nevertheless presented for completeness and posterity.

6. Please correct typos and inconsistencies in the manuscript (some are mentioned in the minor points) because it was difficult for me to keep track.

Apologies for any spelling mistakes. These should now be entirely rectified.

Minor points

7. Line 14-15: author mentions that “E3 retention is mediated by the E3-binding protein (E3BP), which has not previously been clearly resolved within the PDC.” which is not true as it has been previously shown by the author himself, in Kyrilis et al Cell reports 2021 (incorrectly cited), in Tüting et al, and later on in Skalidis et al (not cited here). Proteomic, activity, western blot, and cryo-EM reconstruction data have supported this retention.

What was meant here is that it has not been resolved to this degree by direct reconstruction, but the reviewer's criticism is warranted. This has been rephrased as “*which is here resolved within the PDC core from *N.crassa*, resolved to 3.2Å*”

8. Please revise also claims in Line 45.

In the submitted manuscript, line 45 reads “*Unlike mammals however, fungal E3BP binds to the interior of the E2 core assembly instead of substituting core components(Forsberg et al, 2020; Skalidis et al, 2021; Tüting et al, 2021)(Fig. 1B).*” It is not clear in what way the reviewer disagrees with this, when the preceding critique was that this has long been established. Any claim that the reviewer feels to be exaggerated can be subject to change of course, but this critique should perhaps be clarified.

9. Line 59: the word “either” maybe should be removed or sentence rephrased?

Duly amended

10. Line 73: typo of word “identification”

Duly amended

11. Line 83: residues P420 and L423 are not properly annotated in Figure 2D. maybe the author could change the illustration to show the full interface? or mention that these residues are not shown. The author can also show the complete CBD and its fit in the high-resolution density in a systematic manner, also by rotating the illustration to show the flanking beta strands as well.

Figure 2 has been greatly revised, and these residues are now shown in panel 2F.

12. Line 94: I think numbering is wrong as it should be 347-389 and not 289, please check.

Indeed, it has been duly amended

13. Line 100: the author here stretches a bit the data available as he claims that ONLY M2 is the domain that binds the E2 but the M3 is not clearly resolved thus cannot be excluded as an interacting interface, it would be nice to comment on this, especially when the disordered linker is not included.

The final section of the leading paragraph does allude to this, which has been clarified to read “*M3 might thus be a secondary binding motif. Residual density is observed in the CTD binding pockets where trimeric CBD is sterically impermissible (Fig. S6B), but as it cannot be clearly resolved auxiliary M3 binding should be considered possible but unsubstantiated. Conducted simulations also did not indicate a preference for the M3 region to approach or favorably interact with available binding pockets. The observed density could alternatively be M2 of meta-stable monomeric CBD, that does participate CBD trimers. The universal conservation of M3 thus remains unexplained*”

14. Line 101: typo“an” amphipathic helix... instead of “a”

Duly amended

15. Lines 109-111: very hypothetical in my opinion, please down-tone the claims.

In the submitted manuscript, line 109-111 reads “*One must consider that M3 might act as a secondary binding motif against the CTD binding pocket. Residual density is observed in these interfaces (Fig. S6B), but cannot be clearly resolved. This density could alternatively be meta-stably bound M2 of monomeric CBD that does not form CBD trimers.*” The reviewer pointed out in minor point 13 that M3 might be such a binding motif. Nonetheless, the statement now affirms that “*auxiliary M3 binding should thus be considered possible but unsubstantiated*” in lieu of structured density to support such a notion. The alternative

16. Line 119: Residues K403_CBD-D412-CTD are wrong according to the figure- maybe the author means R425_CBD-D412-CTD?

The additional evidence provided by molecular dynamics does not support these interactions, and they are consequently not mentioned in the main text any longer.

17. Line 125: the author cites Tüting et al, 2021 manuscript here but does not include it in the comparison of eg Figure S6, nor the model organism.

By their own account, Tüting et al constructed an atomic model of the CBD including sidechains, based on a 6.2Å map. They refined this using Phenix, but only provided the PDB as an auxiliary file in an SBgrid database entry. No refinement statistics were provided. While a comparison could be made, the lack of scientific merit in the Tüting model makes such a comparison potentially misleading. Differences that might be attributed to differences between species might simply be due to poor refinement procedure or model statistics. Hence, a direct comparison is not advisable. The model organism used by Tüting et al has however been annotated in Figure S6 (now S8) to place it in the phylogenetic context presented here.

18. Line 144: words “unambiguously” and “throughout” have typos

Duly amended

19. Line 148-149: sentence that starts with “Were...” I think should be rephrased, maybe it is “Where...”?

Duly amended to “*If core-substitution would be possible*”

20. Line 151-152: author mentions that there are some search criteria in Figure S6 that I don't see. These criteria are not clear to me. Fig S6 shows phylogenetic relationship of fungal species

This sentence indicates that the criteria has resulted in the tripartite division, which is shown in fig S6. The phrasing has been changed to clarify this.

21. Figure S2 is a bit difficult for me to understand, there are typos in the legend, and in general can you please optimize it? The distances that are presented are not described in the legend.

Figure S2 (now S6) has been duly amended. The Figure is more clear with colors corresponding to other figures. Distances are mentioned explicitly and the wording has been made more clear.

22. Line 152-154: I think the author here stretches out a bit the available data to make this conclusion, and also reports the obvious of the eukaryotic origin of E3BP.

The shared evolutionary origin of E2 and E3BP has been established prior to the present paper, and any phrasing that might suggest otherwise has been changed to avoid confusion. However, the joint evolutionary origin of E3BP across eukaryotic species has not been established, and is not at all certain from existing literature and evidence. The present work however provides evidence to support this notion. The sentence has however been changed to the following, to assure claims are not exaggerated: *“It therefore seems likely that fungal and animal E3BP are indeed orthologs, with a shared evolutionary origin, and that E3BP is a eukaryotic gene”*.

23. Line 160: word “analyzed” has a typo

Duly amended

24. Line 197: word “specific” typo

Duly amended

25. Line 198: words “with this” mistyped

Duly amended

26. Line 237: is it Fig. S6 or it should be Fig. S7? Please double-check citations to Figures and Tables of the manuscript because I might have missed something.

Duly amended

27. Line 245: “....for the both the...” rephrase

Duly amended

28. Line 263: “Pez M3 is however more similar to Zyg M3 that to its own M2....” Is there a typo here?

Duly amended

29. Line 269: word “Within” has a typo

Duly amended

30. Line 298: ...the PSBD...” space missing

Duly amended

31. Line 307: typo instead of “to”, “co is written

Duly amended

32. Line 315 and 324: Celsius miss the degree symbol?

Duly amended

33. Line 376: “s” missing from word “protein” as should be plural

Duly amended

34. Line 520: delete word “the” after “...hydrophobicity that stabilizes...”

Duly amended

35. Line 530: "...symmetry axis" There is a typo here and further on the same line it is either "additional" or "addition of"

Duly amended

36. Line 531: "from utilizing ny of the remaining 3 binding sites that are most proximal to both G347 and V392." Typo

Duly amended

37. Line 532: typo of word "either"

Duly amended

38. In general in the text and also in Supp. Fig S2 the author claims these residual density in the binding pocket. If it was E3BP then also more densities of the E3BP would be visible????maybe the author could explain if he saw more than this except the disordered parts

The densities are not claimed, they are simply observed. Their origin is discussed in the context, and attributed either to M3 or M2 of monomeric CBDs which do not participate in canonical CBD trimers. Due to the symmetric binding interface, monomeric CBD would present the challenge of a mixture of binding orientations in addition to flexibility, inherent disorder and a small structured domain. The very limited flexibility of the CBD trimer was just enough to be classifiable in the present data, indicating that the increased challenge of extending these residual densities by further analysis is unlikely to be successful. Biochemical approaches would be needed to more confidently attribute these densities to either M2 or M3.

39. Line 573: A, B and C are missing from the Fig. S5 even if mentioned in the legend

This has been amended, and S5 is now S8

40. Line 577: word "if" is a typo

Duly amended

41. Supp Fig S4, please put scale bar in the cryo-EM image

Added, and the figure re-organized as feedback from per reviewer 2.